# Buzz, Choose, Forget: A Meta-Bandit Framework for Bee-Like Decision Making

## Abstract

We introduce a sequential reinforcement learning framework for imitation learning designed to model heterogeneous cognitive strategies in pollinators. Focusing on honeybees, our approach leverages trajectory similarity to capture and forecast behavior across individuals that rely on distinct strategies: some exploiting numerical cues, others drawing on memory, or being influenced by environmental factors such as weather. Through empirical evaluation, we show that state-of-the-art imitation learning methods often fail in this setting: when expert policies shift across memory windows or deviate from optimality, these models overlook both fast and slow learning behaviors and cannot faithfully reproduce key decision patterns. Moreover, they offer limited interpretability, hindering biological insight. Our contribution addresses these challenges by (i) introducing a model that minimizes predictive loss while identifying the effective memory horizon most consistent with behavioral data, and (ii) ensuring full interpretability to enable biologists to analyze underlying decision-making strategies and finally (iii) providing a mathematical framework linking bee policy search with bandit formulations under varying exploration–exploitation dynamics, and releasing a novel dataset of 80 tracked bees observed under diverse weather conditions. This benchmark facilitates research on pollinator cognition and supports ecological governance by improving simulations of insect behavior in agroecosystems. Our findings shed new light on the learning strategies and memory interplay shaping pollinator decision-making.

## 1 Introduction

Over the past decade, researchers have increasingly turned to artificial intelligence (AI) and computational modeling to replicate or simulate animals' decision processes, referred to as imitation learning (Cully et al., 2015). In this case, the goal is to train an agent to learn by observing and reproducing the animal's behavior in the same way as if the animals were experts. In particular, reinforcement learning (RL) frameworks have gained increasing attention as a way to describe how animals learn from trial and error, as an alternative to statistical models or simple heuristic rules. These RL models serve a dual purpose: they help biologists to understand how these animals learn to facilitate rule discovery (Wason, 1960) (i.e. policy modelisation) from real animal data experiments, and they make it possible to run virtual ecological interventions (for instance, simulating how bees that switch between policies would respond to guidance toward pesticide-free zones). However, existing imitation-learning and RL-based models still face major limitations when applied to bees for some reasons: (1) some of them exclude the balance between contextual and non-contextual strategies in the decision process modeling. (2) They overlook the archetypal mechanism of limited-memory learning ; we define here the *memory* of the animal by a parameter, $S$, that truncates the observation history to the $S$ most recent observations. This parameter needs to be optimized in the imitation learning. (3) These models assume homogeneity among bees, although individuals may exhibit distinct behaviors and no explainability is given for each individual. (4) They require access to the full trial sequence and cannot operate online, making them unsuitable for sequential, real-time prediction of behavior. Some bees are able to understand the context information to limit the regret in their strategies, and some others do not (Giurfa et al., 2022). The overarching challenge is therefore to provide a model that can both explain and forecast the policy of each individual bee. This paper proposes a new algorithm to model bees behaviors focusing on contextual binary foraging

tasks (scenarios with two alternatives in a Y-maze, with left vs. right choices, where the reward is systematically located on the side presenting the highest stimulus number, a cue the bee can perceive before making its choice. We summarize key methodologies and show (1) how to identify the best $\tau$ window size that estimates $S$, (2) how individuals vary according to their strategies, and (3) how can we forecast any individual's policy regardless of their specific skills in an online setting. Our method is summarized in Fig. 1. Our code is open-source and our data are openly available.[1].

**A new imitation learning framework**   Imitation learning (IL) enables agents to acquire behavior from expert demonstrations in order to limit costly or unsafe exploration (Zhao et al., 2020). By grounding policy optimization in expert trajectories, IL offers a sample-efficient framework to capture adaptive strategies. However, most IL methods are designed to imitate experts that behave near-optimally, and therefore struggle when the expert exhibits non-optimal, heterogeneous, or temporally shifting policies. This is mostly explained by the fact that they prioritize policy optimization over expert imitation. Unlike classical IL, biological experts (such as bees) often follow non-optimal policies. In nature, several policies may coexist, and a single expert can change its strategy over time. This is largely due to the limited memory of insects, which restricts decisions to a short history of past actions and rewards. Therefore, attention must be paid to how the expert guides the agent, and how the agent adapts to multiple experts. It requires not only defining what should be imitated, but also handling this memory limitation. When data are collected in real time, for instance via drone-based tracking, an online framework becomes indispensable to process and interpret bee behavior as observations arrive. Our contribution is MAYA (Multi Agent Y-maze Allocation), which enables bee imitation learning on a sequential two-choice learning (Y-maze). MAYA combines several multi armed bandit (MAB) policies (including random and contextual variants) with a fixed memory setting $\tau$ for similarity evaluation. Similarity evaluation can be based on probability of success (with Kullback-Leibler or Wasserstein distance) or on trajectory (with Dynamic Time Warping DTW similarity). The best choice of similarity is made according to the ability to imitate the expert and limit the cumulative cost of wrongly replicated actions over trials. Then, our paper studies the similarity that should be used.

**Understanding the learning skill**   MAYA models bee policies as mixtures of multiple MAB agents, thus providing a quantitative framework for characterizing behavioral variability across individuals. Since bees possess limited memory of past experiences, their decision policies may shift over time. Such shifts are not random but reflect different effective strategies depending on the recent learning window $S$. To capture this, MAYA decomposes the observed trajectories into segments that align with distinct agent models, each defined by a specific MAB. These MAB vary according to their strategies: pure exploration, deterministic or stochastic reward-based choice between left and right arms of the Y-maze, and context-dependent strategies where cues guide decisions (see App. 13). By structuring bee behavior as a combination of such MAB, MAYA not only reproduces expert trajectories but also yields an interpretable description of MAB policy shifts and memory constraints.

**Window-size discovering**   MAYA requires the specification of a sliding window $\tau \in \mathcal{T}$ in order estimate $S$ and to select the importance of the past information used to align the behavior of the bee and the MAB. Because memory is a biological constraint rather than a freely tunable parameter, it is essential to determine which $\tau$ best reflects the bee's effective learning horizon $S$. In our experiments, we assess how this setting influences the ability to imitate the bee. We find, for example that the optimal window length decreases under adverse weather conditions, consistent with the idea that environmental noise reduces the usable amount of past information, but generally stabilizes around seven past trials across all datasets. As a complementary analysis, we also include experiments with mice, where a similar optimal setting emerges. We also include simulated data to validate the robustness of MAYA under controlled conditions, where the ground-truth strategy is known.

**Open dataset and ecological insights**   We release to the community a new open dataset recording experiments on 80 bees (with $[22 - 40]$ sequential trials per bee) across 5 diverse situations (favorable and adverse weather, in Oceania and Europe). More details about the experiment are given in App 6.1. In each experiment, a bee enters a Y-maze where it is exposed to a number of visual stimuli presented on both the left and right arms. The reward is consistently located on the side displaying

---

[1]https://anonymous.4open.science/r/maya-4E30

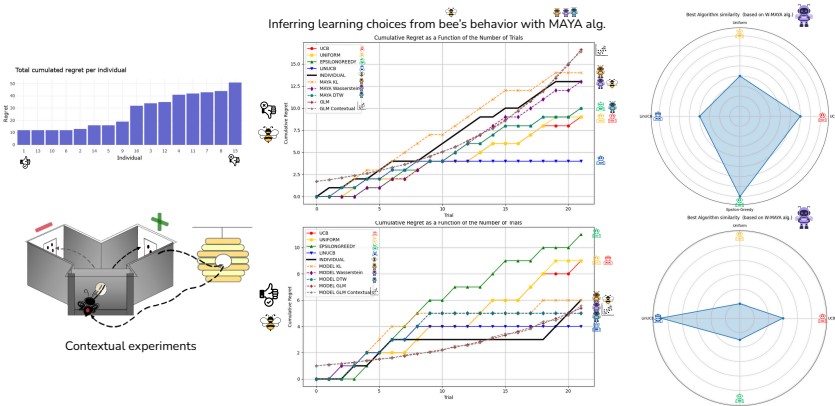

Figure 1: MAYA (Multi-Agent Y-maze Allocation) is an imitation learning framework for policy selection via windowed regret matching. Leveraging logged bee trajectories and three similarity metrics (Wasserstein, KL, DTW), MAYA maps learning dynamics onto 2-armed bandit strategies (UCB, Epsilon-greedy, LinUCB, Uniform). Beyond performance alignment, MAYA provides interpretability of bee behaviors by revealing differences in memory span and learning aptitude, thereby distinguishing "good learners" from "poor learners" in contextual experiments.

the greater number of stimuli. During a session, each bee performs between 22 (for one dataset) and 40 (for the other datasets) trials, where the number of stimuli (and therefore the rewarded side) is randomly assigned at each trial.

## 2 PRELIMINARIES

**Problem formulation.** We model the bee prediction task (forecast the decision left or right) as an RL problem. At each trial $t \in 1, \ldots, T$ the environment reveals a state $s_t \in \mathcal{S}$ described by the number of trial $t$ and available contextual information : $x_t \in \mathbb{R}^2$ with $x_t = $ (stimuli on Left and Right side, weather, ...). The bee selects an action (i.e. chose a side) : $a_t \in \mathcal{A} := \{L, R\}$, corresponding to Left and Right. Then, the bee receives a reward $r_t = r(s_t, a_t) \in \{0, 1\}$, which captures whether the choice is correct or incorrect (e.g., sugar or quinine). This model is actually a Markov Decision Process (MDP) (Sutton & Barto, 2018). It is defined as a tuple $(\mathcal{S}, \mathcal{A}, P, \mathcal{R})$ with a state space $\mathcal{S}$, an action space $\mathcal{A}$. In our setting, $S = |\mathcal{S}|$ and $A = |\mathcal{A}|$ are finite (i.e $S, A < \infty$). The quantity $P = (P_a : a \in \mathcal{A})$ is called the transition function with $P_a : \mathcal{S} \times \mathcal{S} \to [0, 1]$ and so $P_a(s, s')$ is the probability that the agent moves from state $s$ in state $s'$ according to action $a$. The set space $\mathcal{R}$ is defined by all outputs of reward functions $r_a$ according an action $a : \mathcal{R} = (r_a : a \in \mathcal{A})$. We are on a discrete-time series system such as the initial state is defined by $S_1$. In each round $t$ the agent observes the state $S_t \in \mathcal{S}$, chooses an action $A_t \in \mathcal{A}$ and receives the reward $r_{A_t}(S_t)$. The environment then samples $S_{t+1}$ from the probability vector $P_{A_t}(S_t) \in P$. The **history** $H_t = (S_1, A_1, r(S_1, A_1), \ldots, S_{t-1}, A_{t-1}, r(S_{t-1}, A_{t-1}), S_t)$ or more simply $H_t = (S_1, A_1, r_1, \ldots, S_{t-1}, A_{t-1}, r_{t-1}, S_t)$, contains the information available before the action for the round $t$ is to be chosen. A **policy** is a (possibly randomised) map from the set of possible histories to actions. The set of such policies is denoted by $\Pi$ and its elements are identified with maps $\pi : \mathcal{A} \times \mathcal{S} \to [0, 1]$ with $\sum_{a \in \mathcal{A}} \pi(a|s) = 1$ for any $s \in \mathcal{S}$ so that $\pi(a|s)$ is interpreted as the probability that policy $\pi$ takes action $a$ in state $s$. We are on a finite-trial experiment i.e. $t \in \{1, \ldots, T\}$ where $T$ is the total number of trials. We consider here that all rewards are equivalent whatever the future, then an **optimal policy** $\pi^*$ for a discrete time $T$ system is a policy that satisfies, for any state $s : \pi^* = \arg\max_{\pi \in \Pi} \sum_{t=1}^{T} \gamma^t r(A_t, S_t)$ with $\gamma = 1$. Finally, let $N_t(a)$ denote the total number of times action $a$ has been selected up to round $t$. We define $Q_t(a) = \frac{1}{N_t(a)} \sum_{j=1}^{t-1} r_j \mathbf{1}_{\{a_j = a\}}$ as the simple average of rewards which have been observed.

**Regret.** Let $\pi^\star$ denote the (unknown) optimal policy. The *instantaneous regret* at trial $t$ is defined as $\Delta_t = r(s_t, a_t^\star) - r(s_t, a_t)$, where $a_t^\star := \pi^\star(s_t) = \arg\max_{a \in \mathcal{A}} r(s_t, a)$ is the optimal action

under the state $s_t$. The *cumulative simple regret* after $T$ trials is the sum of instantaneous regrets $R(\pi, 1, T) = \sum_{t=1}^{T} \Delta_{\pi,t}$.

In the experiment, the reward given to the bee at each state $s_t$ does not depend on the state $s_{t-1}$. Hence our model can be seen as a 2-armed bandit problem and not a classical reinforcement learning problem. Bees differ in solving such learning task. Based on biology literature (Capela et al., 2024), their different behaviors can be modeled by four different two-armed bandit strategies (MAB) :

1. **Epsilon Greedy** (Sutton & Barto, 1998) : exploits current action that maximize observed average reward (i.e. .) and explore the other action according a small probability ($\epsilon$).

$$A_t = \left\{ \begin{array}{l} \text{Argmax}_a[Q_t(a)] \text{ with probability } 1 - \epsilon \\ a \sim \text{Uniform}(\mathcal{A} \backslash \{\text{Argmax}_a[Q_t(a)]\}) \text{ with probability } \epsilon \end{array} \right.$$

2. **Optimistic stategy UCB style** (Auer et al., 2002) : construct an adaptative upper confidence bound around $Q_t(a)$ . In this case UCB1 chose according :

$$A_t = \text{Argmax}_a[Q_t(a) + \sqrt{\frac{\ln t}{N_t(a)}}].$$

The number of trials on $N_a(t)$ and empirical observed reward on each arms are considered.

3. **Contextual-multi-armed bandits (CMAB) LINUCB style** (Li et al., 2010) : At the beginning of trial $t$, the agent observes a context $x_t$. It's redefine the choice of an action according the context information $x_t$. Let $G_a = \boldsymbol{X}_a^\top \boldsymbol{X}_a + \lambda \boldsymbol{I}$ where $\boldsymbol{X}_a$ is the matrix with the context vectors of action $a$ as rows, $\boldsymbol{I}$ the identity matrix and $\lambda \in \mathbb{R}$ is a regularization parameter. LINUCB1 chose according :

$$A_t = \text{Argmax}_a[\, x_t^\top \hat{\Theta}_{a,t} + \sqrt{x_t^\top G_a^{-1} x_t}\,].$$

where $\hat{\Theta}_{a,t} \in \mathbb{R}^2$ are estimated parameter of action $a$ at $t$.

4. **Random choice strategy UNIFORM style** : At each trial, the agent chooses an action uniformly at random, independently of past observations or contexts. This baseline strategy does not exploit reward or contextual informations, and serves as a comparison.

$$A_t \sim \text{Uniform}(\mathcal{A}).$$

Among the strategies considered above, LINUCB1 (ref as LINUCB) is the only bandit algorithm here that explicitly incorporates contextual information. Consequently, it is the sole approach capable of asymptotically converging to the optimal policy in our Y-maze experimental setting. Regardless of its ability to adopt the optimal strategy (i.e., to use contextual information), the bee selects an action $A_t$ based on memory history. This memory reflects the history of past actions, rewards, and contexts. However, learning and memory of honeybees can be impacted by a large amount environmental conditions, like the weather variation (Gérard et al., 2022). Additionally, the learning process in itself may be reflected by the succession of sub-optimal strategies (based on $Q_a(t)$ or based on a random choice) to the optimal strategy (based on the contextual information) with potential transitive states. Therefore, comparing bee strategies with these four policies must be carried out in a **non-stationary** framework. Unfortunately, the effective history length is difficult to anticipate, as it may evolve in many ways (Fiandri et al., 2024).

Then, to incorporate this *bee's memory* concept, defined in psychology as the recency effect (Glanzer & Cunitz, 1966), we introduce the concept of a sliding window $\tau \in \mathcal{T}$ to lay the stress on recent history. The history is restricted to $H_{t,\tau} = (S_{t-\tau}, A_{t-\tau}, r_{t-\tau}, \ldots, S_{t-1}, A_{t-1}, r_{t-1}, S_t)$ and the policy becomes $\pi : \mathcal{A} \times \mathcal{S} \times \mathcal{T} \to [0, 1]$ with $\sum_{a \in \mathcal{A}} \pi(a|s, \tau) = 1$. The simple regret according to $\tau$ is : $R(\pi, \tau, 1, T) = \sum_{t=\tau}^{T} \Delta_{\pi,t}$.

**Imitation learning to approximate a bee's behaviour**   Our goal is to learn a policy $\pi_{\text{MAYA}}$ which is close to $\pi_{\text{bee}}$. The selection of the best MAB algorithm that mimics a bee's behaviour is based on comparing at $t$ the $\tau$-last cumulative regret trajectories generated $(R(\pi, \tau, 1, t))$ by the bee and by each candidate MAB. For this, for a well-chosen similarity distance $d$. We define for two policies $\pi_1$ and $\pi_2$, their distance according to $\tau$ and $t$ trials: $\delta(\pi_1, \pi_2, \tau, t) := d(R(\pi_1, \tau, 1, t), R(\pi_2, \tau, 1, t))$.

In the following, we consider three choices for $d$, reflecting two complementary interpretations of the regret sequence. First, when regrets are viewed as random variables, we use distributional distances such as Kullback–Leibler (KL) divergence and Wasserstein distance, which measure similarity in probabilistic structure or geometric displacement. Second, when regrets are interpreted as a temporal trajectory, we use Dynamic Time Warping (DTW), which emphasizes temporal alignment and is robust to local timing fluctuations.

Finally, the success of the imitation learning algorithm will be quantified here using the following cost of a wrong **reproduced action**. Let :

$$c(s_t|a_t) = \left\{ \begin{array}{l} 1 \text{ if } a_t \neq \pi_{\text{bee}}(s_t) \\ 0 \text{ otherwise} \end{array} \right.$$

Assume that $\pi_{\text{MAYA}}(a \neq \pi_{\text{bee}}(s)|s) \leq \varepsilon$, with $\varepsilon \in [0,1]$ then (Ross et al., 2010) shows that $\mathbb{E}[\sum_{t=1}^{T} c(s_t, a_t)] \leq \varepsilon T$. If $\pi_\theta$ is learned by minimizing previous distances, success is measured by considering this cost. See App. 16 for additional details on this metric.

## 3 CONTRIBUTION

Our contribution, the MAYA algorithm, addresses the challenge of inverse reinforcement learning in biology when expert demonstrations are heterogeneous, non-stationnary and not necessarily optimal. Instead of assuming a single coherent expert policy, MAYA explicitly treats bee trajectories as mixtures of potentially distinct and sometimes sub-optimal MAB strategies. By dynamically aligning the observed behaviour with a set of candidate MAB policies, MAYA captures both successful learning episodes and non-optimal or inconsistent actions, which are common in insect cognition. We present here a condensed version of the MAYA framework; the complete algorithmic description is provided in Appendix 11.

**Inputs.**    The algorithm takes as input the logged regret trajectory of a bee policy $R(\pi_{\text{bee}}, 1, T)$, a finite set $\mathcal{P} = \{\pi_1, \ldots, \pi_N\}$ of $N$ candidate bandit policies, and a window size $\tau$. The window size controls how much historical regret information is used at each step: for $t < \tau$ the algorithm uses all past data, while for $t \geq \tau$ it only considers the most recent $\tau$ steps.

**Initialization.**    The algorithm initializes a placeholder policy $\pi_\theta$ and an agent buffer $\xi$.

**Warm-up Phase** ($t < \tau$).    For each time step $t \in \{2, \ldots, \tau - 1\}$:

1. We define $\tau = t - 1$ and the algorithm observes the bee regret $R(\pi_{\text{bee}}, \tau, 1, t-1)$ with the context information $x_t$.

2. For each candidate policy $\pi_i \in \mathcal{P}$, the algorithm simulates its action distribution $\pi_i(s_{t-1}|x_t)$ and computes the cumulative regret $R(\pi_i, \tau, 1, t-1)$.

3. A distance $d(\cdot, \cdot)$ is computed between the bee regret trajectory and the simulated regret of $\pi_i$, then we compute $\xi_t = \operatorname{argmin}_{\pi \in \mathcal{P}} \delta(\pi_{\text{bee}}, \pi, t)$ according to the choice of $d(.)$. In case of a tie, $\xi_t$ is sampled from the set of best candidates.

4. The algorithm updates $\pi_\theta$ to imitate $\pi_{\xi_t}$, i.e. $\pi_\theta(a_t|s_{t-1}) \leftarrow \pi_{\xi_t}(a_t|s_{t-1})$, and we store $\xi[t] \leftarrow \xi_t$.

The chosen policy $\pi_\theta$ is then used to sample the next action $A_t$, a reward $r_t$ is received, and all candidate policies are updated.

**Windowed Phase** ($t \geq \tau$).    For subsequent steps $t \in \{\tau, \ldots, T\}$, the procedure is analogous, except that we fix $\tau$ as a hyperparameter. Then, only the most recent observations $\tau$ are used when computing regret and $\xi_t = \operatorname{argmin}_{\pi \in \mathcal{P}} \delta(\pi_{\text{bee}}, \pi, \tau, t)$. Specifically, regret and policy regrets are evaluated over the interval $[t - \tau, t - 1]$ rather than the full trajectory. Again, the best match $\xi_t$ is calculated between each policy $\pi \in \mathcal{P}$, and $\pi_\theta$ is updated according to the best match.

**Output.**    After $T$ steps, the algorithm returns the policy $\pi_{\text{MAYA}} = \pi_\theta$, which best matches the bee's regret profile, while adapting online to the context and rewards.

## 3.1 SIMILARITY EVALUATION

The algorithm depends on the choice of the distance $d$ between the trajectories of the regrets. We will consider three distinct distances. For a review of different distances, see for instance in (Besse et al., 2015).

1. Dynamic Time Warping (DTW). One of the most used similarity measures between two paths is given by the so-called DTW. It is defined as follows. Given two temporal sequences $X = (x_1, \ldots, x_{T_1})$ and $Y = (y_1, \ldots, y_{T_2})$ over $E \subset \mathbb{R}^d$ with $d \in \mathbb{N}^*$. DTW aligns them by finding an admissible path $\psi = \{(i_k, j_k)\}_{k=1}^K$ that respects temporal ordering. Formally, the DTW is defined as $DTW(X, Y) = \min_\psi \sum_{k=1}^K \|x_{i_k} - y_{j_k}\|$, with $K \in \mathbb{N}^*$ where the minimization runs over all monotone alignment paths $\psi$ between the indices of $X$ and $Y$. This distance enables comparison of sequences with different lengths or temporal distortions, by optimally stretching or compressing the time axis.

2. KL-distance. In this case, the sequence of the regrets is considered as a realization at each step of a Bernoulli distribution. Hence we can define the Kullback-Leibler distance between each trajectory by a proper normalization. Set $Q$ a probability measure on $E$. If $P$ is another probability measure on $(E, \mathcal{B}(E))$, then the KL divergence is $D_{\mathrm{KL}}(P\|Q) = \int_E \log \frac{\mathrm{d}P}{\mathrm{d}Q} \mathrm{d}P$, if $P \ll Q$ and $\log \frac{\mathrm{d}P}{\mathrm{d}Q} \in L^1(P)$, and $+\infty$ otherwise.

3. Wasserstein-distance. We consider again the distributional point of view. The 1-Wasserstein distance is defined as follows. For two distributions $\pi_1$ and $\pi_2$ over $E \subset \mathbb{R}^d$ a compact subset, endowed with the norm $\|.\|$, recall that their 1-Wasserstein distance is defined as $W_1(\pi_1, \pi_2) = \min_{\pi \in \Pi(\pi_1, \pi_2)} \int_{x \in E, y \in E} \| x - y \| \, d\pi(x, y)$, where $\Pi(\pi_1, \pi_2)$ denotes the set of distributions on $E \times E$ with marginals $\pi_1$ and $\pi_2$.

## 3.2 THEORETICAL ANALYSIS

We provide in App 14 worst-case upper bounds on the cumulative regret gap between $\pi_{\mathrm{MAYA}}$ and $\pi_{\mathrm{bee}}$ across stationary and $S$ cyclic regimes, expressed in terms of $T$, $\tau$ and $S$. We inform the choice of $\tau$ to control the error in non-stationary settings. In App;15, we extended our experimental protocol to include 42 simulated datasets, resulting in more than 100.800 synthetic trajectories. Overall, these analyses validate the theoretical justification of our $\tau$-range and demonstrate its empirical robustness across diverse $S$ switching regimes. We also added a small grid search over exploration parameters : $\epsilon \in \{0.1, 0.2, 0.3\}$, and $\alpha_{\mathrm{ucb}}, \alpha_{\mathrm{linucb}} \in \{0.5, 1, 1.5, 2, 4\}$.

## 4 EXPERIMENTAL EVALUATION

Our experiments aim to address the following questions: i/ What is the best window size ($\tau$) to estimate $S$ and similarity metric (DTW, KL, Wass) to approximate bee learning? ii/ What information can MAYA provide about the exploratory and contextual process of bees? iii/ How external information (here, the weather) can impact the window size parameter $\tau$?

**Experiment description** The datasets vary according to location (three from France and two from Australia) and weather conditions (two cold, one moderate, and two hot). Each dataset contains the trajectories of 16 bees with 22 or 40 trials (depending on the dataset, see App 6.1 for more details). We also include a complementary experiment in the App 12, adapted from (Ashwood et al., 2020b), using data from mice performing perceptual decision-making tasks. We complete our study with simulated data in App 15.

**Metrics** MAYA and comparative methods are evaluated based on their ability to minimize the cost of incorrectly reproduced actions over the sequence of trials. We then report, for our five datasets, the $MSE\left(\sum_{t=1}^T c(s_t \mid a_t)\right)$ and $MAE\left(\sum_{t=1}^T c(s_t \mid a_t)\right)$, computed across all bees within the same dataset. We first observe how these metrics evolve in Sec.4.1. We further include a variance-based residual statistical test comparing the bee's cumulative regret with MAYA. We provide in App.16 more explanation with a numeric toy example. Then, in Sec. 4.2, we show how MAYA generates

trajectories that closely match those of bees across all datasets. Sec 4.3 provides an example of individual analysis. We also observe how trajectories are clustered in a similar manner in Sec.4.4. This can be considered as an additional performance metric: the ability to assign trajectories to the same cluster.

## 4.1 BEST WINDOW SIZE AND DISTANCE METRICS

Figure 2 reports the average MSE and MAE results according to $\tau$ for the five datasets. When several MAB agents are the best candidates for $\xi_t$, the selection is random, which can introduce some variability. We report average MSE/MAE over $\tau \in [3, \min(T, 30)]$ and over the full-history value $\tau = T$. We also provide the standard deviation for several $\tau$ in the App 6.2. However, the MAYA MSE/MAE recorded standard deviations are small and nearly constant. This is easily explained: if agents follow the same action sequence, their costs are identical. Therefore, the effect of randomness is limited. This can be seen in Fig.3, for example, where UCB and UNIFORM act identically at the beginning of the experiment.

Across all datasets, the results confirm the trend that for $\tau \in [5, 10]$ the losses decrease. However, weather influences the optimal $\tau$. Weather conditions modulate this optimum. Cold weather requires to choose $\tau \in [5, 7]$, moderate weather $\tau \in [6, 8]$, and hot weather $\tau \in [7, 10]$. Based on this observation, we set $\tau = 7$ as a robust compromise across all conditions. This observation is similar in complementary experiments with mice (App 12). Then, we fix $\tau = 7$ for the rest of the paper. Whatever $\tau$, MAYA–Wass provides the best results across all datasets.

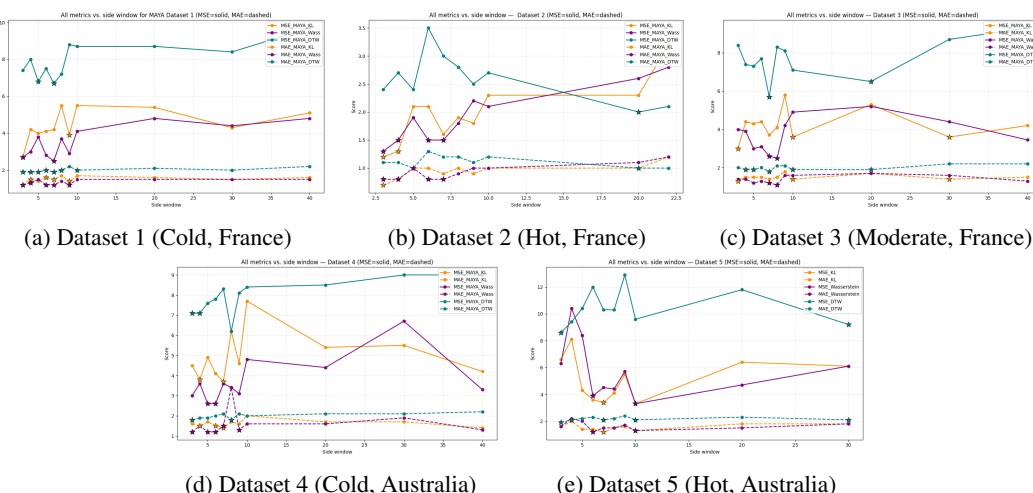

(a) Dataset 1 (Cold, France)   (b) Dataset 2 (Hot, France)   (c) Dataset 3 (Moderate, France)

(d) Dataset 4 (Cold, Australia)   (e) Dataset 5 (Hot, Australia)

Figure 2: Comparative study of the best window size $\tau$ by average MSE and MAE; weather and location for each dataset are provided. The maximum window value corresponds to using the full sequence (i.e., no window). $\star$ symbol refers as best performance according standard deviation and average reward (see Tab9. in App 10 for the full results)

### 4.1.1 STATISTICAL TEST: VARIANCE-BASED RESIDUAL ANALYSIS

To quantify how well a model reproduces the behaviour of an individual bee, we analyse the *residual trajectory* : $e_t = R(\pi_{\text{bee}}, 1, t) - R(\pi_{\text{model}}, 1, t)$, where $R(\pi, 1, t)$ denotes cumulative simple regret without $\tau$ restriction. Because regret trajectories are monotone and strongly autocorrelated, we focus on the *variance* of the residuals rather than their absolute level. Let $s_e^2 = \text{Var}(e_t)$, and $s_{\text{bee}}^2 = \text{Var}(R(\pi_{\text{bee}}, 1, t))$. Under the null hypothesis : $H_0 : s_e^2 \leq s_{\text{bee}}^2$, the model's deviations are no larger than the intrinsic variability of the animal. We compute the Fisher statistic : $F_{\text{obs}} = \frac{s_e^2}{s_{\text{bee}}^2}$, and obtain a one-sided $p$-value $p = \text{Pr}(F \geq F_{\text{obs}} \mid H_0)$. Small $p$-values indicate that the model fails to capture the individual's dynamics; large values indicate a good match. Table 1 reports (mean, min, max) $p$-values per dataset.

Overall, MAYA-based models (MAYA-KL, MAYA-Wass, MAYA-DTW) achieve the highest minimal $p$-values across all datasets, whereas baseline bandit algorithms (UCB, LinUCB, Uniform, $\epsilon$-greedy) exhibit larger residual dispersion and more frequent rejection of $H_0$. The results also show that, in Dataset 2, many bees display a clear LinUCB-like phase, whereas in the other datasets the dominant patterns are UCB-, EpsilonGreedy or Uniform-like. The reported minimum $p$-value for each dataset reflects whether *at least one* bee deviates significantly from the model; values below 0.05 lead to rejecting $H_0$. MAYA-KL and MAYA-Wass consistently achieve near-perfect alignment, and MAYA-DTW performs similarly except for a single bee in Dataset 2, likely due to the sensitivity of DTW to constant cumulative regret trajectories, which are more frequent in this dataset. We also include a new WORST baseline that always selects the suboptimal arm; the opposite BEST baseline is equivalent to LinUCB and is therefore omitted.

| Algorithm | dataset1 | | | dataset2 | | | dataset3 | | | dataset4 | | | dataset5 | | |
|---|---|---|---|---|---|---|---|---|---|---|---|---|---|---|---|
| | mean | min | max | mean | min | max | mean | min | max | mean | min | max | mean | min | max |
| UCB | 0.79 | 0.01 | 1 | 0.85 | 0.01 | 1 | 0.75 | 0.01 | 1 | 0.82 | 0.01 | 1 | 0.91 | 0.24 | 1 |
| LINUCB | 0.63 | 0.55 | 0.72 | 0.90 | 0.60 | 0.99 | 0.64 | 0.57 | 0.72 | 0.61 | 0.51 | 0.83 | 0.62 | 0.54 | 0.73 |
| UNIFORM | 0.91 | 0.01 | 1 | 0.82 | 0.04 | 1 | 0.89 | 0.01 | 1 | 0.91 | 0.01 | 1 | 0.94 | 0.65 | 1 |
| E-GREEDY | 0.91 | 0.01 | 1 | 0.74 | 0.01 | 1 | 0.91 | 0.01 | 1 | 0.93 | 0.01 | 1 | 0.95 | 0.80 | 1 |
| WORST | 0.34 | 0.01 | 0.99 | 0.28 | 0.01 | 0.99 | 0.29 | 0.01 | 0.99 | 0.33 | 0.01 | 0.99 | 0.28 | 0.01 | 0.97 |
| MAYA-KL | **0.99** | **0.99** | 1 | **0.92** | **0.65** | 1 | **0.99** | **0.99** | 1 | **1.00** | **0.99** | 1 | **0.99** | **0.94** | 1 |
| MAYA-Wass | **1.00** | **1.00** | 1 | **0.99** | **0.91** | 1 | **1.00** | **1.00** | 1 | **1.00** | **1.00** | 1 | **0.99** | **0.99** | 1 |
| MAYA-DTW | **0.93** | **0.01** | 1 | **0.90** | **0.01** | 1 | **0.97** | **0.80** | 1 | **0.99** | **0.87** | 1 | **0.99** | **0.97** | 1 |

Table 1: Performance of all algorithms across five real bee datasets (16 bees per dataset), reported in terms of (mean, min, max) Fisher-test (one side) $p$-values per dataset. Lower values indicate a worse match, while values near 1 indicate high similarity to the subject's trajectory. MAYA consistently achieves the highest $p$-values across datasets, indicating superior trajectory alignment. The code is available on our GitHub repository.

## 4.2 COMPARATIVE STUDY OF REPRODUCTIVE BEHAVIOR

We compare the performance of MAYA-Wasserstein, MAYA-KL and MAYA-DTW with all IRL algorithms implemented in the imitation library of (Gleave et al., 2022). It includes implementations of Generative Adversarial Imitation Learning (GAIL), Behavioral Cloning (BC), Dataset Aggregation (DAgger), Adversarial Inverse Reinforcement Learning (AIRL), Density-based reward modeling (DBR), Reward Learning through Preference Comparisons (Pref-Comp), Maximum Causal Entropy Inverse Reinforcement Learning (MCE) and Soft Q Imitation learning (SQIL). These methods are the baseline references of IRL methods. We provide a full explanation of these methods in App 8. We also compare our results with a generalized linear model (GLM) applied to the full trajectory. In this case, the GLM captures each bee trajectory through a response transformation, while allowing the variance of each measurement to depend on its predicted value. We further introduce a variant that incorporates contextual information $x_t$ as covariates (GLM-Context).

The reported results are in Tab 2 for the MSE. As the best performances are almost identical for the MAE we provide MAE results in App 8.1. Methods such as Pref-comp, MCE, and DBR tend to overshoot the bee trajectories and focus mainly on minimizing regret (policy optimization), as the context provides all the necessary information to choose correctly. These methods fail to reproduce bee behavior: the divergence between the cumulative regret trajectories grows over time, since the learned policy accumulates substantially less regret. In fact, these methods generally act like LinUCB. GAIL, Dagger and SQIL fail to capture the full range of behaviors, instead tending to mimic the most frequently represented populations in the dataset. AIRL reproduces bee trajectories identically, and can therefore be seen as a full copy-paste of the dataset without any real capacity for generalization (see App. 8.1 for more details). AIRL actually memorizes observed trajectories rather than capturing the underlying decision-making mechanisms. When we fine-tune the parameters of these methods, we reinforce the influence of the expert on the learning process (see App 9). However, this requires more computation time, and the resulting MSE/MAE values are higher than those obtained with MAYA (considering a large set of $\tau$). The GLM can be considered the most challenging baseline to outperform in terms of MSE/MAE, since it is explicitly designed to fit the bee's $T$-trial trajectory rather than to learn a policy. Adding contextual covariates has only a minor influence, which is expected because the GLM primarily captures direct statistical dependencies

Table 2: MSE comparison of methods across the five datasets. Values are reported as mean $\pm$ standard deviation. We fix $\tau = 7$ for all MAYA variant. Best performance of comparative methods are reported here.

| Dataset | GAIL | BC | AIRL | Dagger | DBR | MCE | Pref-Comp | SQIL | GLM (no ctx) | GLM (ctx) | MAYA-KL | MAYA-Wass | MAYA-DTW |
|---|---|---|---|---|---|---|---|---|---|---|---|---|---|
| 1 | 29.6±41 | **5.16±3** | **0±0** | 22.8±32 | 43.1±54 | 148.83±38 | 104±57 | 26.2±19 | **3.0±1** | **3.0±1** | 4.2±3 | **2.5±1** | 6.7±7 |
| 2 | 23.2±17 | **2.86±2** | **0±0** | 9.67±12 | 15.26±16 | 49.5±14 | 24.54±18 | 9.8±6 | **1.4±2** | **1.4±2** | 1.6±1 | 1.5±1 | 1.3±3 |
| 3 | 27.5±40 | **5.5±4** | **0±0** | 21.6±46 | 41.38±51 | 140.3±34 | 125.7±44 | 22.6±15 | **3.1±1** | **3.1±1** | 3.7±3 | **2.6±1** | 5.7±5 |
| 4 | 25.3±39 | **5.35±4** | **0±0** | 22.9±34 | 46.06±55 | 148.2±39 | 124.1±52 | 25.3±20 | **3.0±1** | **3.0±1** | 3.7±3 | **3.6±2** | 8.3±10 |
| 5 | 47.7±45 | 26.7±42 | **0±0** | 25.8±47 | 115.7±242 | 374±311 | 284 +/-254 | **25.0±16** | 8.0±8 | **7.9±8** | 3.4±3 | 4.5±5 | 10.3±11 |

rather than adaptive decision-making. Among all variants, MAYA with the Wasserstein distance (MAYA-Wass) consistently achieves the best performance across datasets, highlighting its robustness in capturing trajectory similarity. We report in Fig 3a and Fig3b MAYA's fitting for two bees.

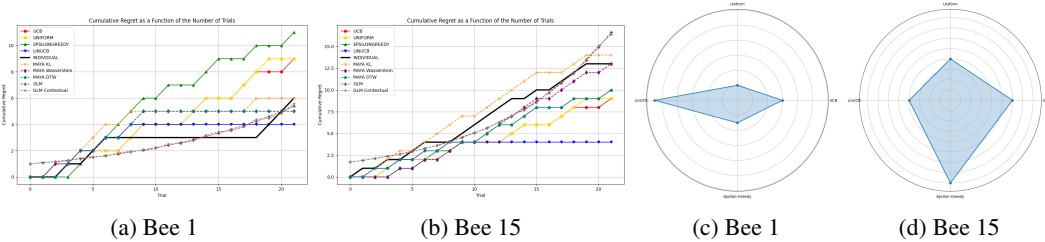

| (a) Bee 1 | (b) Bee 15 | (c) Bee 1 | (d) Bee 15 |

Figure 3: For bee 1 (fast learner, low regret) and bee 15 (slow learner, high regret) from dataset 2. **Left**: Cumulative regret of 4 MABs, GLM and MAYA ($\tau = 7$) for two bees from Dataset 2. **Right**: Choice interpretability with MAYA–Wass ($\tau = 7$), Left: LinUCB, Top: Uniform, Right: UCB, Bottom: EpsilonGreedy.

### 4.3 UNDERSTANDING THE LEARNING PROCESS

MAYA provides post-hoc behavioral explainability at both the individual-bee and dataset levels. Concretely, our explainability metric is the alignment rate: the proportion of time MAYA's chosen action $a_t$ matches the action prescribed by a reference MAB policy (e.g., LinUCB, UCB, Epsilon-greedy, or Uniform). Empirically, when focusing on low-regret bees, MAYA's decisions align predominantly with LinUCB-like choices (see Fig3c). In contrast, for high-regret bees, decisions align more often with Epsilon greedy (see Fig3d) as a context-agnostic heuristic tends to over-exploit early, locking onto arms that were temporarily lucky and thus yielding higher cumulative regret (exploited 80% of the time). Table 3b reports the MAB-policy alignment proportions aggregated across all bees from the five datasets. Across MAYA variants, alignment profiles are broadly similar, though MAYA–DTW displays greater variability, likely due to its sensitivity to local misalignments and abrupt strategy shifts.. The strong performance of MAYA–Wass appears consistent with its higher alignment to LinUCB-style trajectories, suggesting better identification of context-dependent decision patterns. Additional per-$\tau$ and per-dataset analyses are provided in App 7. There we also show that using a smaller temporal window ($\tau \leq 5$) tend to produce unstable alignment patterns, especially for slow learners, whose behavior requires longer horizons to disambiguate. These empirical proportions are useful for biologists as priors for forward ecological simulations: one can generate realistic behavioural sequences by sampling actions from the inferred mixture over reference policies.

### 4.4 SIMULATE AND FORECAST THE BEE TRAJECTORY

We study MAYA's ability to forecast realistic trajectories that capture bee behavior. We consider two behavioral types (slow and fast learners) defined by their cumulative regret according to the number of trials. We jointly cluster real bee trajectories and MAYA-generated trajectories separately, and we assess alignment by checking whether real and simulated samples fall into the same clusters via a confusion matrix. We define a clustering function $\kappa$ that assigns each trajectory $R(.)$ to a cluster: $\kappa : R(.) \mapsto \{1, \ldots, K\}$, where $K$ is the number of clusters. As an additional performance evaluation, we compute the proportion of cases where the model and bee trajectories fall into the

same cluster: $\text{ClusterAcc} = \frac{1}{J} \sum_{i=1}^{J} \mathbf{1}\left[ \kappa(R(\pi_{\text{bee}}^j, 1, T) = R(\pi_{\text{MAYA}}^j, 1, T) \right]$, where $J$ denotes the total number of bees across all datasets (here, 80), $\pi_{\text{bee}}^j$ is the policy of the $j$th bee, and $\pi_{\text{MAYA}}^j$ is the policy generated by MAYA for the $j$th bee. We use $K = 2$ clusters to mirror the two archetypes. Clustering I (Euclidean) uses Euclidean distance on time series and requires equal lengths. We pool all bees across datasets. Lengths differ, so we truncate each series to the minimum common length (22). We apply a second clustering with Dynamic Barycenter Averaging (DBA); a DTW-based clustering method. It handles unequal lengths and local time shifts, so no truncation is needed. We report DBA clustering of real bee trajectories in App10. We report the confusion matrix (real vs. simulated labels) in Tab 3a. The MAYA-Wass error fluctuations remain bounded with Gaussian amplitude. It shows that MAYA-Wass's dynamics are stable, almost 0-centered with a maximum standard deviation error equal to 3. We provide additional figures of Euclidean Clustering in App 10.

Table 3: **Left**: For all bees in the five datasets, we report average ClusterAcc (%) under two prototype aggregation regimes: (i) Euclidean averaging with a maximum sequence length of 22, and (ii) DBA with a maximum sequence length of 40. Across both regimes, MAYA–Wass achieves the highest accuracy (79% and 91%), followed by *MAYA–KL* and *MAYA–DTW*. Standard errors are $\leq 1\%$ for all entries and are omitted for readability. **Right**: Proportion of $a_t$ according to all trials for all dataset (5). We fix $\tau = 7$ for all MAYA variants.

|  | MAYA-KL | MAYA-Wass | MAYA-DTW |
|---|---|---|---|
| ClusterAcc (Euclidean, Max L = 22) | 77% | 79% | 70% |
| ClusterAcc (DBA, Max L = 40) | 84% | 91% | 80% |

(a) ClusterAcc (%)

|  | Epsilon-Greedy | Lin-UCB | UCB | Uniform |
|---|---|---|---|---|
| MAYA-KL | 34.4%±2 | 10.5%±1 | 22.6%±1 | 32.5%±2 |
| MAYA-W | 31.1%±1.5 | 16.2%±0.8 | 22.2%±0.9 | 30.5%±1.4 |
| MAYA-DTW | 36.5%±2.5 | 10.8%±1 | 17.5%±1 | 35.2%±2.3 |

(b) MAYA explainability for all bees choices

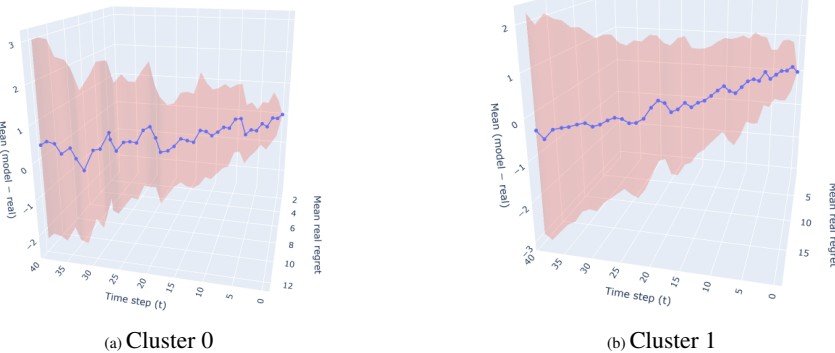

(a) Cluster 0              (b) Cluster 1

Figure 4: Average difference between MAYA-Wass ($\tau = 7$) predictions and real trajectories, expressed as $R(\pi_{\text{MAYA-Wass}}, 1, t) - R(\pi_{\text{bee}}, 1, t)$ (z-axis), for the two DBA clusters (0 and 1). The red band represents $\pm\sigma$ (standard deviation). These surfaces show how the average prediction error evolves across trials ($t$) and cumulative regret (y-axis).

## 5 DISCUSSION

We introduced MAYA, a sequential imitation-learning model that forecasts individual bee trajectories across heterogeneous cognitive strategies. Across datasets and weather conditions, a memory window of ($\tau = 7$) represents a reasonable trade-off with weather-driven variability while maintaining robust predictive performance. It corresponds to 15–30 minutes in our protocol ($\approx$ seven trials, depending on the bee). Among variants, MAYA-Wass achieved the strongest overall performance, while MAYA-KL and MAYA-DTW remained competitive. Beyond accuracy, MAYA provides interpretable, per-trial explanations of choice, enables the generation of "artificial bees," and supports forward simulation for ecological what-if scenarios. These results position MAYA as a viable alternative to IRL baselines and traditional statistical models. Future work will deploy MAYA in large-scale ecological simulations to assess its predictive value for ecological management decisions.

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

# 6 APPENDIX

## 6.1 DATASET DESCRIPTION

In this dataset, bees are confronted to a numerical discrimination task. Bees first enter the maze in an entrance chamber before flying through a hole and facing two images located at the end of each arm. The image has a different number of dots : for example in dataset 1 and 2, one of the image has two dots while the other have four dots. If the bee chooses the correct image (i.e. the side with the highest number of dots), it will be rewarded with a sugar reward (50% sugar/water) placed in a pipette in the middle of the image, alternatively if it chooses the incorrect image, then it will be punished by finding a bitter tasting solution (quinine solution) within the pipette. Bees cannot detect (neither visually nor by odor) which solution is located where. Then, they are only able to know the image on each side before choosing. Between each trials the bee will go back to the hive to deliver the collected sugar, before returning back the maze for another trial (typically lasting a few minutes). During this time, the experimenter randomly changes the images or not, and varying the position of the dots. The localization of the correct image alternate between the right and left arm according to a pseudo-random sequence. Each dataset include 16 bees.

Table 4: Datasets summary

| Dataset | nb indiv | $T$ | Location | Weather |
|---------|----------|-----|----------|---------|
| Dataset 1 | 16 | 40 | France | Cold |
| Dataset 2 | 16 | 22 | France | Hot |
| Dataset 3 | 16 | 40 | France | Moderate |
| Dataset 4 | 16 | 40 | Australia | Cold |
| Dataset 5 | 16 | 30 | Australia | Hot |

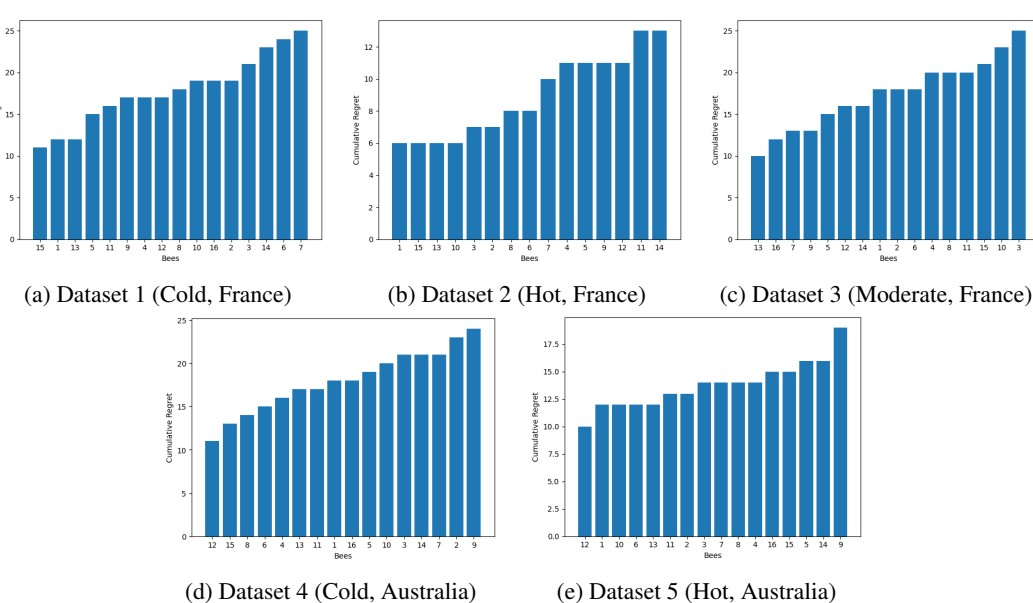

(a) Dataset 1 (Cold, France)     (b) Dataset 2 (Hot, France)     (c) Dataset 3 (Moderate, France)

(d) Dataset 4 (Cold, Australia)     (e) Dataset 5 (Hot, Australia)

Figure 5: Proportion of cumulative regret for the five datasets, per bees

## 6.2 MSE AND MAE OF MAYA ACCORDING TO τ

| side_window | MAYA_KL mean MSE | MAYA_KL mean MAE | MAYA_Wass mean MSE | MAYA_Wass mean MAE | MAYA_DTW mean MSE | MAYA_DTW mean MAE |
|---|---|---|---|---|---|---|
| 3.0 | 2.7±2 | 1.2±0.7 | 2.7±2 | 1.2±0.6 | 7.4±10 | 1.9±1 |
| 4.0 | 4.2±4 | 1.5±0.8 | 3.0±2 | 1.3±0.6 | 8.0±13 | 1.9±1 |
| 5.0 | 4.0±4 | 1.4±0.9 | 3.8±3 | 1.5±0.6 | 6.8±7 | 1.9±0.9 |
| 6.0 | 4.1±2 | 1.6±0.6 | 2.8±2 | 1.2±0.5 | 7.5±7 | 2.0±1 |
| 7.0 | 4.2±3 | 1.5±0.7 | 2.5±1 | 1.2±0.5 | 6.7±7 | 1.9±1 |
| 8.0 | 5.5±5 | 1.7±0.9 | 3.7±3 | 1.4±0.7 | 7.2±7.8 | 2.0±1 |
| 9.0 | 3.9±3 | 1.4±0.7 | 2.9±2 | 1.2±0.6 | 8.8±9 | 2.2±1 |
| 10.0 | 5.5±5 | 1.7±0.9 | 4.1±4 | 1.5±0.8 | 8.7±10 | 2.0±1 |
| 20.0 | 5.4±5 | 1.6±0.8 | 4.8±5 | 1.5±0.9 | 8.7±10 | 2.1±1 |
| 30.0 | 4.3±3 | 1.5±0.6 | 4.4±3 | 1.5±0.7 | 8.4±10 | 2.0±1 |
| $T=40$ | 5.1±5 | 1.6±1 | 4.8±6 | 1.5±0.9 | 9.7±11 | 2.2±1 |

Table 5: Dataset 1 (Cold weather, France)

| side_window | MAYA_KL mean | MAYA_KL mean | MAYA_Wass mean | MAYA_Wass mean | MAYA_DTW mean | MAYA_DTW mean |
|---|---|---|---|---|---|---|
| 3.0 | 1.2±1 | 0.7±0.4 | 1.3±1 | 0.8±0.4 | 2.4±1 | 1.1±0.4 |
| 4.0 | 1.3±0.8 | 0.8±0.3 | 1.5±1 | 0.8±0.4 | 2.7±2 | 1.1±0.6 |
| 5.0 | 2.1±1 | 1.0±0.4 | 1.9±2 | 1.0±0.5 | 2.4±2.7 | 1.0±0.6 |
| 6.0 | 2.1±1 | 1.0±0.5 | 1.5±1 | 0.8±0.4 | 3.5±3 | 1.3±0.7 |
| 7.0 | 1.6±1 | 0.9±0.4 | 1.5±1 | 0.8±0.4 | 3.0±3 | 1.2±0.6 |
| 8.0 | 1.9±1 | 1.0±0.3 | 1.8±1 | 0.9±0.4 | 2.8±2 | 1.2±0.6 |
| 9.0 | 1.8±1 | 0.9±0.4 | 2.2±2 | 1.0±0.6 | 2.5±2 | 1.1±0.6 |
| 10.0 | 2.3±2 | 1.0±0.5 | 2.1±2 | 1.0±0.6 | 2.7±1 | 1.2±0.6 |
| 20.0 | 2.3±1 | 1.0±0.4 | 2.6±1 | 1.1±0.3 | 2.0±1 | 1.0±0.4 |
| $T=22$ | 3.2±3 | 1.2±0.6 | 2.8±1 | 1.2±0.4 | 2.1±1 | 1.0±0.5 |

Table 6: Dataset 2 (Hot weather, France)

| side_window | MAYA_KL mean MSE | MAYA_KL mean MAE | MAYA_Wass mean MSE | MAYA_Wass mean MAE | MAYA_DTW mean MSE | MAYA_DTW mean MAE |
|---|---|---|---|---|---|---|
| 3.0 | 3.0±2 | 1.3±0.6 | 4.0±4 | 1.4±0.8 | 8.4±12 | 2.0±1.4 |
| 4.0 | 4.4±4 | 1.5±0.9 | 3.9±4 | 1.4±0.8 | 7.4±11 | 1.9±1 |
| 5.0 | 4.3±4 | 1.5±0.7 | 3.0±3 | 1.2±0.7 | 7.3±11 | 1.9±1 |
| 6.0 | 4.4±4 | 1.5±0.8 | 3.1±2 | 1.3±0.6 | 7.7±9 | 2.0±1 |
| 7.0 | 3.7±3 | 1.4±0.6 | 2.6±1 | 1.2±0.5 | 5.7±5 | 1.8±0.8 |
| 8.0 | 4.1±3 | 1.5±0.7 | 2.5±1 | 1.1±0.4 | 8.3±9 | 2.1±1 |
| 9.0 | 5.8±5 | 1.8±0.8 | 4.2±2 | 1.6±0.6 | 8.1±8 | 2.1±1 |
| 10.0 | 3.6±3 | 1.4±0.7 | 4.9±5 | 1.6±1 | 7.1±9 | 1.9±1 |
| 20.0 | 5.3±4 | 1.7±0.8 | 5.2±5 | 1.7±0.7 | 6.5±8 | 1.9±1 |
| 30.0 | 3.6±2 | 1.4±0.5 | 4.4±3 | 1.6±0.7 | 8.7±9 | 2.2±1 |
| $T=40$ | 4.2±4 | 1.5±0.8 | 3.45±3 | 1.3±0.6 | 9.3±11 | 2.2±1 |

Table 7: Dataset 3 (Moderate weather, France)

| side_window | MAYA_KL mean | MAYA_KL mean | MAYA_Wass mean | MAYA_Wass mean | MAYA_DTW mean | MAYA_DTW mean |
|---|---|---|---|---|---|---|
| 3.0 | 4.5±4 | 1.6±0.9 | 3.0±4 | 1.2±0.9 | 7.1±10 | 1.8±1.3 |
| 4.0 | 3.8±3 | 1.5±0.7 | 3.6±3 | 1.5±0.6 | 7.1±9 | 1.9±1 |
| 5.0 | 4.9±3 | 1.7±0.7 | 2.6±3 | 1.2±0.7 | 7.6±11 | 1.9±1 |
| 6.0 | 4.1±3 | 1.5±0.7 | 2.6±1 | 1.2±0.4 | 7.8±9 | 2.0±1 |
| 7.0 | 3.7±3 | 1.4±0.6 | 3.6±2 | 1.5±0.5 | 8.3±10 | 2.1±1 |
| 8.0 | 6.2±8 | 1.7±1 | 3.4±2 | 3.4±2 | 6.2±7 | 1.8±1 |
| 9.0 | 4.6±3 | 1.6±0.7 | 3.1±2 | 1.3±0.5 | 8.1±7 | 2.1±1 |
| 10.0 | 7.7±7 | 2.0±1 | 4.8±4 | 1.6±0.8 | 8.4±10 | 2.0±1 |
| 20.0 | 5.4±4 | 1.7±0.8 | 4.4±2 | 1.6±0.5 | 8.5±11 | 2.1±1.2 |
| 30.0 | 5.5±4 | 1.7±0.7 | 6.7±7 | 1.9±0.9 | 9.0±12 | 2.1±1 |
| $T=40$ | 4.2±5 | 1.4±0.8 | 3.3±2 | 1.3±0.6 | 9.0±10 | 2.2±1 |

Table 8: Dataset 4 (Cold weather, Australia)

| side_window | MAYA_KL mean MSE | MAYA_KL mean MAE | MAYA_Wass mean MSE | MAYA_Wass mean MAE | MAYA_DTW mean MSE | MAYA_DTW mean MAE |
|---|---|---|---|---|---|---|
| 3 | 6.6±9 | 1.6±1 | 6.3±9 | 1.6±1 | 8.6±10 | 1.9±1 |
| 4 | 8.1±8 | 2.0±1 | 10.4±12 | 2.2±1 | 9.4±8 | 2.1±1 |
| 5 | 4.3±5 | 1.4±0.9 | 8.4±10 | 2.0±1 | 10.4±12 | 2.2±1 |
| 6 | 3.6±3 | 1.4±0.7 | 3.9±8 | 1.2±1 | 12.0±11 | 2.3±1 |
| 7 | 3.4±3 | 1.2±0.9 | 4.5±5 | 1.5±1 | 10.3±11 | 2.1±1 |
| 8 | 4.1±3 | 1.5±0.6 | 4.4±5 | 1.5±0.9 | 10.3±12 | 2.2±1 |
| 9 | 5.5±8 | 1.6±1 | 5.7±6 | 1.7±1 | 12.9±16 | 2.4±1 |
| 10 | 3.3±3 | 1.3±0.6 | 3.3±3 | 1.3±0.7 | 9.6±10 | 2.1±1 |
| 20 | 6.4±6 | 1.8±1 | 4.7±5 | 1.5±0.8 | 11.8±13 | 2.3±1 |
| $T=30$ | 6.1±5 | 1.8±0.9 | 6.1±5 | 1.8±0.9 | 9.2±10 | 2.1±1 |

Table 9: Dataset 5 (Hot weather, Australia)

Table 10: MSE and MAE of MAYA as a function of the window size $\tau$. The $T$ row denotes the no-window setting ($\tau = T$), where at each trial the full trajectory up to time $t$ is used.

# 7 UNDERSTANDING THE LEARNING PROCESS

## 7.1 MAYA EXPLAINABILITY WITH $\tau = 7$

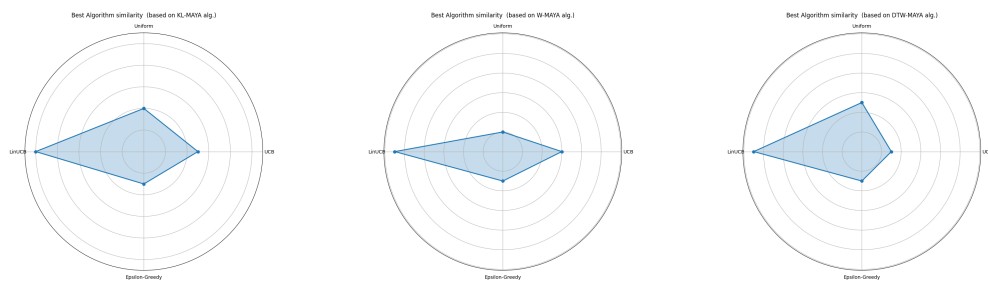

Figure 6: MAYA-KL        Figure 7: MAYA-Wass        Figure 8: MAYA-DTW

Figure 9: For bee 1 (fast learner, low regret) from dataset 2 we report choice interpretability for MAYA-variants ($\tau = 7$). Left: LinUCB, Top: Uniform, Right: UCB, Bottom: EpsilonGreedy.

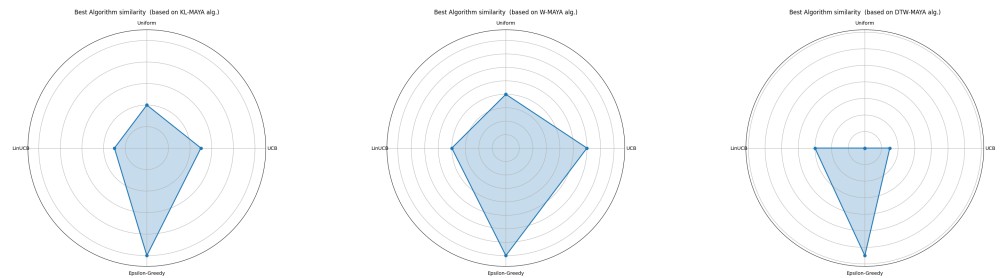

Figure 10: MAYA-KL        Figure 11: MAYA-Wass        Figure 12: MAYA-DTW

Figure 13: For bee 15 (slow learner, high regret) from dataset 2 we report choice interpretability for MAYA-variants ($\tau = 7$). Left: LinUCB, Top: Uniform, Right: UCB, Bottom: EpsilonGreedy.

## 7.2 MAYA EXPLAINABILITY WITH $\tau = 3$

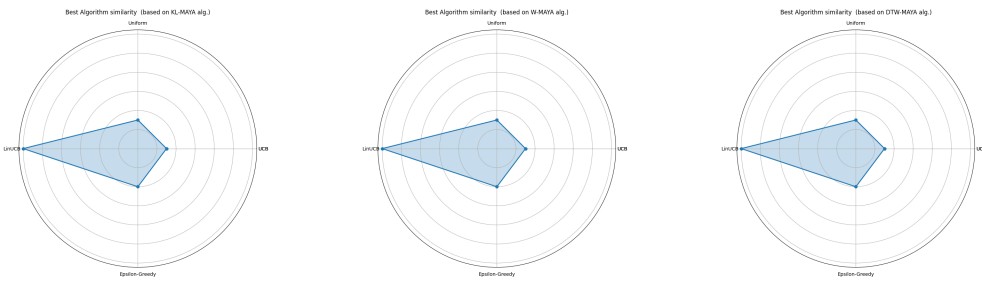

Figure 14: MAYA-KL          Figure 15: MAYA-Wass          Figure 16: MAYA-DTW

Figure 17: For bee 1 (fast learner, low regret) from dataset 2 we report choice interpretability for MAYA-variants ($\tau = 3$). Left: LinUCB, Top: Uniform, Right: UCB, Bottom: EpsilonGreedy.

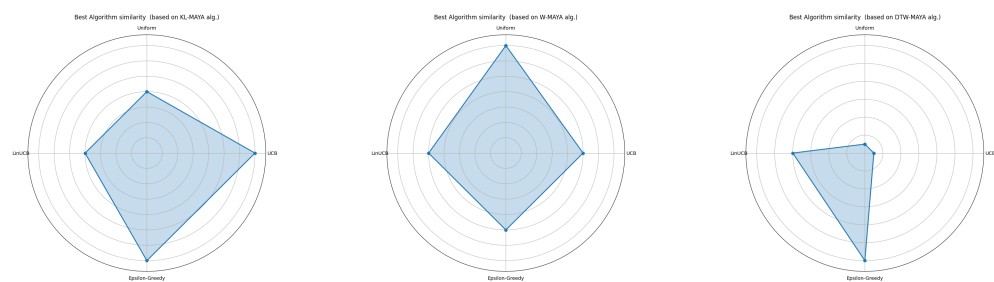

Figure 18: MAYA-KL          Figure 19: MAYA-Wass          Figure 20: MAYA-DTW

Figure 21: For bee 15 (slow learner, high regret) from Dataset 2 we report choice interpretability for MAYA-variants ($\tau = 3$). Left: LinUCB, Top: Uniform, Right: UCB, Bottom: EpsilonGreedy.

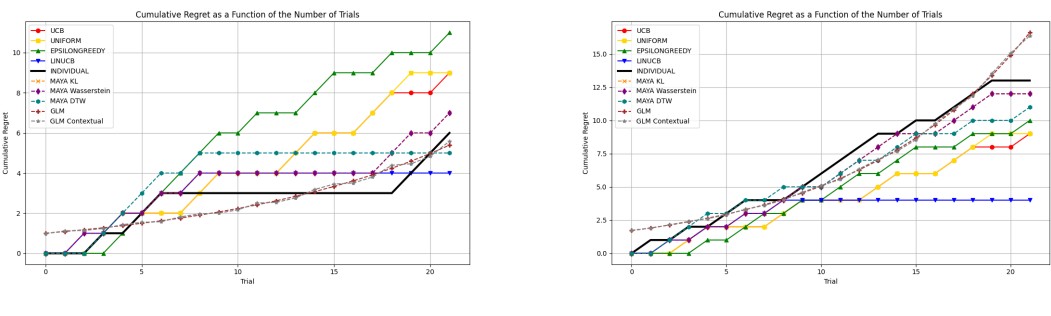

Figure 22: Bee 1          Figure 23: Bee 15

Figure 24: Regret modelization for bee 1 (lower cumulative regret) and bee 15 (higher cumulative regret) of Dataset 2, with $\tau = 3$.

## 8 COMPARATIVE METHODS DESCRIPTION

- Generative Adversarial Imitation Learning (GAIL) GAIL learns a policy by simultaneously training it with a discriminator that aims to distinguish expert trajectories against trajectories from the learned policy. (Ho & Ermon, 2016)

- Behavioral Cloning (BC) Behavioral cloning directly learns a policy by using supervised learning on observation-action pairs from expert demonstrations. It is a simple approach to learning a policy, but the policy often generalizes poorly and does not recover well from errors. (Foster et al., 2024).

- AIRL, similar to GAIL, adversarially trains a policy against a discriminator that aims to distinguish the expert demonstrations from the learned policy. Unlike GAIL, AIRL recovers a reward function that is more generalizable to changes in environment dynamics. (Fu et al., 2018).

- DAgger (Dataset Aggregation) iteratively trains a policy using supervised learning on a dataset of observation-action pairs from expert demonstrations (like behavioral cloning), runs the policy to gather observations, queries the expert for good actions on those observations, and adds the newly labeled observations to the dataset. DAgger improves on behavioral cloning by training on a dataset that better resembles the observations the trained policy is likely to encounter, but it requires querying the expert online (Ross et al., 2011).

- Density-based reward modeling is an inverse reinforcement learning (IRL) technique that assigns higher rewards to states or state-action pairs that occur more frequently in an expert's demonstrations. The key intuition behind this method is to incentivize the agent to take actions that resemble the expert's actions in similar states (Dumoulin et al., 2024).

- Maximum Causal Entropy Inverse Reinforcement Learning (MCE IRL) : The principle of maximum causal entropy is a method that extends the classical maximum entropy idea to sequential settings. Instead of considering probabilities in isolation, it uses causally conditioned probabilities, which means that the model explicitly accounts for the fact that information is revealed step by step over time. This allows us to properly capture how side information becomes available and how it influences decisions at each stage (Biernaskie et al., 2009).

- Preference Comparisons : The preference comparison algorithm learns a reward function from preferences between pairs of trajectories. The comparisons are modeled as being generated from a Bradley-Terry (or Boltzmann rational) model, where the probability of preferring trajectory A over B is proportional to the exponential of the difference between the return of trajectory A minus B. In other words, the difference in returns forms a logit for a binary classification problem, and accordingly the reward function is trained using a cross-entropy loss to predict the preference comparison. (Christiano et al., 2023).

- Soft Q Imitation Learning (SQIL) : Soft Q Imitation learning learns to imitate a policy from demonstrations by using the DQN algorithm with modified rewards. During each policy update, half of the batch is sampled from the demonstrations and half is sampled from the environment. Expert demonstrations are assigned a reward of 1, and the environment is assigned a reward of 0. This encourages the policy to imitate the demonstrations, and to simultaneously avoid states not seen in the demonstrations (Reddy et al., 2020).

- GLM : A Generalized Linear Model (GLM) is a statistical framework that extends linear regression to response variables with non-Gaussian distributions. In our setting, the regret trajectory $R(\pi, 1, T)$ is modeled as a function of time, $R(\pi, 1, T) \sim f(t)$, where $f$ is linked to a linear predictor through a canonical link function. A Poisson GLM is employed when the noise structure is count-like, while a Gamma GLM is used to capture multiplicative noise. This allows us to statistically frame the evolution of regret as a stochastic process, while accounting for heterogeneous variability across agents. (Nelder & Wedderburn, 1972).

- Contextual GLM : The contextual variant incorporates side information (e.g., environmental or experimental conditions) into the predictor, enabling the model to capture how context modulates regret dynamics. Then $R(\pi, 1, T) \sim f(t, x_t)$ (McCullagh & Nelder, 1989).

## 8.1 MAE COMPARISON OF METHODS

Table 11: MAE comparison of methods across the five datasets. Values are reported as mean $\pm$ standard deviation. We fix $\tau = 7$ for all MAYA variant

| Dataset | GAIL | BC | AIRL | Dagger | DBR | MCE | Pref-Comp | SQIL | GLM (no ctx) | GLM (ctx) | MAYA-KL | MAYA-Wass | MAYA-DTW |
|---|---|---|---|---|---|---|---|---|---|---|---|---|---|
| 1 | $3.75 \pm 2.5$ | $\mathbf{1.61 \pm 0.79}$ | $\mathbf{0 \pm 0}$ | $2.9 \pm 2.8$ | $4.3 \pm 3.8$ | $10.38 \pm 1.60$ | $8.35 \pm 3.25$ | $3.71 \pm 1$ | $\mathbf{1.4 \pm 0.3}$ | $\mathbf{1.4 \pm 0.3}$ | $1.5 \pm 0.7$ | $\mathbf{1.2 \pm 0.5}$ | $1.9 \pm 1$ |
| 2 | $3.69 \pm 1.8$ | $\mathbf{1.24 \pm 0.72}$ | $\mathbf{0 \pm 0}$ | $1.93 \pm 1.7$ | $2.72 \pm 1.89$ | $6.04 \pm 1.0$ | $3.7 \pm 1.9$ | $2.18 \pm 0.9$ | $\mathbf{0.8 \pm 0.5}$ | $\mathbf{0.8 \pm 0.5}$ | $1.4 \pm 0.6$ | $1.5 \pm 0.5$ | $2.1 \pm 1$ |
| 3 | $3.62 \pm 2.4$ | $\mathbf{1.79 \pm 0.98}$ | $\mathbf{0 \pm 0}$ | $2.6 \pm 3.1$ | $3.4 \pm 4.1$ | $8.13 \pm 1.10$ | $9.76 \pm 1.75$ | $3.2 \pm 1$ | $\mathbf{1.4 \pm 0.4}$ | $\mathbf{1.4 \pm 0.4}$ | $3.7 \pm 3$ | $\mathbf{2.6 \pm 1}$ | $1.8 \pm 0.8$ |
| 4 | $3.1 \pm 2.8$ | $\mathbf{1.65 \pm 0.86}$ | $\mathbf{0 \pm 0}$ | $3.0 \pm 2.7$ | $4.60 \pm 4.8$ | $10 \pm 1.6$ | $9.7 \pm 1.7$ | $3.2 \pm 1$ | $\mathbf{2.1 \pm 1}$ | $\mathbf{2.1 \pm 1}$ | $1.4 \pm 0.6$ | $1.5 \pm 0.5$ | $2.1 \pm 1$ |
| 5 | $4.9 \pm 2.8$ | $\mathbf{3.23 \pm 3}$ | $\mathbf{0 \pm 0}$ | $6.5 \pm 5.1$ | $5.5 \pm 7.8$ | $15.0 \pm 7.6$ | $14.3 \pm 6.92$ | $4.52 \pm 2$ | $8.0 \pm 8$ | $\mathbf{2.2 \pm 1}$ | $\mathbf{1.2 \pm 0.9}$ | $1.3 \pm 0.7$ | $2.1 \pm 1$ |

**More details about the 0-MSE/MAE of AIRL** AIRL is not guaranteed to reproduce expert trajectories in general (e.g., continuous control), but it can do so in small, deterministic MDPs where the expert policy is simple and near-deterministic. This is exactly our setting: the animal's decisions form a low-dimensional bandit strategy with discrete actions with sequences of 40 trials and where the reward is fully deterministic (according to the door with the highest number of stimuli on the Y-maze). In such environments, AIRL can recovers a reward sequence whose optimal policy is identical to the expert's mapping from states to actions, leading to trajectories that match exactly.

## 9 FINETUNING IMITATION LEARNING

We present ablations over the fine-tuning budget of the IRL methods. As the tuning knobs differ across methods, we use the unified notation $b$ for the method-specific budget (see Tab 12). The best results are summarized in the main text.

| $b^{(GAIL)}$ | $b^{(BC)}$ | $b^{(Dagger)}$ | $b^{(DBR)}$ | $b^{(MCE)}$ | $b^{(PrefComp)}$ | $b^{(PrefComp)}$ |
|---|---|---|---|---|---|---|
| *epochs* | *epochs* | *env. steps* | *epochs* | *epochs* | *# envs* | *eval episodes* |

Table 12: Hyperparameters of each comparative methods.

| | MSE (b=1) | MAE (b=1) | MSE (b=10) | MAE (b=10) | MSE (b=50) | MAE (b=50) |
|---|---|---|---|---|---|---|
| GAIL | 29.6 +/- 41 | 3.75+/-2.5 | 29.6 +/- 41 | 3.75+/-2.5 | 29.6 +/- 41 | 3.75+/-2.5 |
| BC | 23.2 +/- 30.8 | 3.26 +/- 2.74 | 19.8 +/- 26.5 | 3.1+/-2.3 | 5.16+/-3.94 | 1.61+/-0.79 |
| AIRL | 0 +/- 0 | 0 +/- 0 | 0 +/- 0 | 0 +/- 0 | 0 +/- 0 | 0 +/- 0 |
| Dagger | 22.8+/- 32.9 | 2.9+/-2.8 | 36.9+/-52.0 | 3.7 +/- 3.8 | 32.5 +/- 50.6 | 3.7+/- 3.3 |
| Density based reward | 43.1 +/- 54.81 | 4.3+/-3.8 | 43.1 +/- 54.8 | 4.3+/-3.8 | 43.1 +/- 54.8 | 4.3+/-3.8 |
| MCE | 148.83 +/- 38.47 | 10.38 +/- 1.60 | 148.83 +/- 38.47 | 10.38 +/- 1.60 | 148.83 +/- 38.47 | 10.38 +/- 1.60 |
| Pref-Comp | 120.25 +/- 52.1 | 9.17 +/- 2.99 | 114 +/- 53 | 8.9 +/- 2.9 | 104.5 +/- 57 | 8.35 +/- 3.25 |
| SQIL | 26.2 +/-19 | 3.75 +/- 1 | 26.2 +/-19 | 3.75 +/- 1 | 26.2 +/-19 | 3.75 +/- 1 |

Table 13: Dataset 1 (Cold weather, France)

| | MSE (b=1) | MAE (b=1) | MSE (b=10) | MAE (b=10) | MSE (b=50) | MAE (b=50) |
|---|---|---|---|---|---|---|
| GAIL | 23.2 +/- 17 | 3.69 +/- 1.8 | 23.2 +/- 17 | 3.69 +/- 1.8 | 23.2 +/- 17 | 3.69 +/- 1.8 |
| BC | 12.1+/-12.1 | 2.54+/-1.74 | 7.3+/-7.7 | 1.99+/-1.3 | 2.86 +/- 2.95 | 1.24 +/- 0.72 |
| AIRL | 0 | 0 +/- 0 | 0 +/- 0 | 0 +/- 0 | 0 +/- 0 | 0 +/- 0 |
| Dagger | 15.63 +/- 19.2 | 2.54+/-2.2 | 11.8 +/- 16.5 | 2.1+/-2.0 | 9.67+/- 12.6 | 1.93 +/-1.7 |
| Density based reward | 15.26 +/- 16.43 | 2.72 +/- 1.89 | 15.26 +/- 16.43 | 2.72 +/- 1.89 | 15.26 +/- 16.43 | 2.72 +/- 1.89 |
| MCE | 49.5 +/- 14.2 | 6.04 +/- 1.0 | 49.5 +/- 14.2 | 6.04 +/- 1.0 | 49.5 +/- 14.2 | 6.04 +/- 1.0 |
| Pref-Comp | 24.54+/-18.3 | 3.7 +/-1.9 | 30.15 +/-17.3 | 4.49 +/- 1.53 | 28.84 +/- 16.13 | 4.46 +/- 1.30 |
| SQIL | 9.80 +/-6 | 2.18+/-0.9 | 9.80 +/-6 | 2.18+/-0.9 | 9.80 +/-6 | 2.18+/-0.9 |

Table 14: Dataset 2 (Hot weather, France)

|  | MSE (b=1) | MAE (b=1) | MSE (b=10) | MAE (b=10) | MSE (b=50) | MAE (b=50) |
|---|---|---|---|---|---|---|
| GAIL | 27.5 +/- 40 | 3.62 +/-2.5 | 27.5 +/- 40 | 3.62 +/-2.5 | 27.5 +/- 40 | 3.62 +/-2.5 |
| BC | 15.9+/-24 | 2.67 +/- 2.26 | 22.0+/-25 | 3.55+/-2.1 | 5.5+/-4.1 | 1.79+/-0.98 |
| AIRL | 0 +/- 0 | 0 +/- 0 | 0 +/- 0 | 0 +/- 0 | 0 +/- 0 | 0 +/- 0 |
| Dagger | 35.4+/- 61.8 | 3.3 +/-3.7 | 34.5+/-48.2 | 3.5 +/- 3.4 | 21.6 +/-46.0 | 2.6 +/-3.1 |
| Density based reward | 41.38 +/- 51.1 | 3.4+/-4.1 | 41.38 +/- 51.1 | 3.4+/-4.1 | 41.38 +/- 51.1 | 3.4+/-4.1 |
| MCE | 140.3 +/-34.7 | 8.13 +/-1.10 | 140.3 +/-34.7 | 8.13 +/-1.10 | 140.3 +/-34.7 | 8.13 +/-1.10 |
| Pref-Comp | 130.98 +/-44.7 | 9.98 +/-1.98 | 134.12+/-37 | 10.12 +/-1.39 | 125.70 +/- 44.1 | 9.76 +/- 1.75 |
| SQIL | 22.65+/-15 | 3.2+/-1 | 22.65+/-15 | 3.2+/-1 | 22.65+/-15 | 3.2+/-1 |

Table 15: Dataset 3 (Moderate weather, France)

|  | MSE (b=1) | MAE (b=1) | MSE (b=10) | MAE (b=10) | MSE (b=50) | MAE (b=50) |
|---|---|---|---|---|---|---|
| GAIL | 25.3 +/-39 | 3.1 +/- 2.8 | 25.3 +/-39 | 3.1 +/- 2.8 | 25.3 +/-39 | 3.1 +/- 2.8 |
| BC | 23.2 +/- 28.6 | 3.4+/-2.4 | 22.3 +/- 26.1 | 3.5 +/-2.2 | 5.35+/-4.17 | 1.65 +/-0.86 |
| AIRL | 0+/-0 | 0+/-0 | 0+/-0 | 0+/-0 | 0+/-0 | 0+/-0 |
| Dagger | 22.9 +/-34.0 | 3.0 +/- 2.7 | 45.3 +/- 52.8 | 4.6 +/- 3.6 | 24.4 +/- 24.2 | 3.2 +/- 2.7 |
| Density based reward | 46.06 +/-55 | 4.60+/-4.8 | 46.06 +/-55 | 4.60+/-4.8 | 46.06 +/-55 | 4.60+/-4.8 |
| MCE | 148.2 +/- 39.6 | 10.3 +/-1.6 | 148.2 +/- 39.6 | 10.3 +/-1.6 | 148.2 +/- 39.6 | 10.3 +/-1.6 |
| Pref-Comp | 124.1 +/-52 | 9.4 +/- 2.78 | 128.29 +/- 42.7 | 9.86 +/- 1.68 | 125.68 +/- 44.19 | 9.7 +/- 1.7 |
| SQIL | 25.3 +/-20 | 3.2 +/- 1 | 25.3 +/-20 | 3.2 +/- 1 | 25.3 +/-20 | 3.2 +/- 1 |

Table 16: Dataset 4 (Cold weather, Australia)

|  | MSE (b=1) | MAE (b=1) | MSE (b=10) | MAE (b=10) | MSE (b=50) | MAE (b=50) |
|---|---|---|---|---|---|---|
| GAIL | 45.71 +/- 45.7 | 4.9 +/- 2.8 | 45.71 +/- 45.7 | 4.9 +/- 2.8 | 45.71 +/- 45.7 | 4.9 +/- 2.8 |
| BC | 124.4 +/- 186.46 | 6.94 +/- 7.05 | 39.7+/- 70 | 3.91+/-3 | 26.7+/-42.7 | 3.23 +/- 3.17 |
| AIRL | 0+/-0 | 0+/-0 | 0+/-0 | 0+/-0 | 0+/-0 | 0+/-0 |
| Dagger | 113.4 +/-247.5 | 6.0+/-7.1 | 93.2 +/-115.9 | 6.5 +/- 5.1 | 25.8 +/- 47.1 | 6.5 +/- 5.1 |
| Density based reward | 115.7 +/- 242.51 | 5.5 +/-7.8 | 115.7 +/- 242.51 | 5.5 +/-7.8 | 115.7 +/- 242.51 | 5.5 +/-7.8 |
| MCE | 374 +/-311.9 | 15.0+/-7.6 | 374 +/-311.9 | 15.0+/-7.6 | 374 +/-311.9 | 15.0+/-7.6 |
| Pref-Comp | 284 +/-254 | 12.9 +/- 7 | 335.6 +/-271 | 14.5 +/- | 332.8 +/- 272.29 | 14.3 +/-6.92 |
| SQIL | 25 +/- 16 | 4.52+/-2 | 25 +/- 16 | 4.52+/-2 | 25 +/- 16 | 4.52+/-2 |

Table 17: Dataset 5 (Hot weather, Australia)

## 10 CLUSTERING

With DBA-clustering, the shift of the simulated centroids toward lower cumulative regret values is explained by the structure of our datasets: one of the five datasets contains bees with trajectories of only 22 trials. When aggregated with 40-trial trajectories, these short sequences lower the average cumulative regret in DBA-based clustering, which pulls the corresponding centroid downward. This effect is expected, since DBA aligns sequences globally and is sensitive to systematic differences in trajectory length.

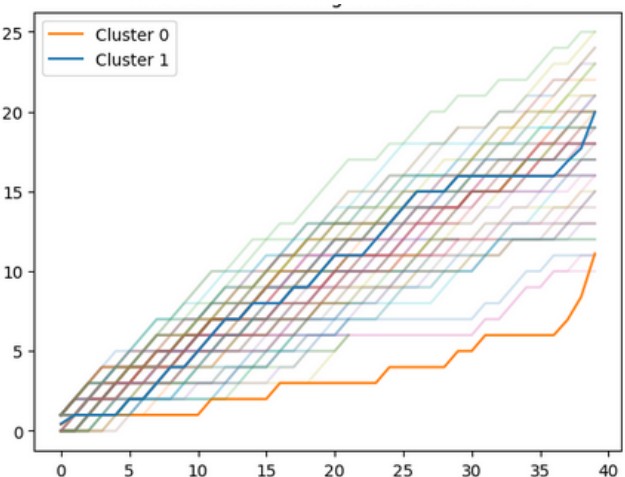

Figure 25: Real bee trajectories clustered into two groups using DBA-based $k$-means. Each curve represents the cumulative regret trajectory of one of the 80 bees, and the two centroid trajectories summarize the dominant behavioural modes observed in the dataset.

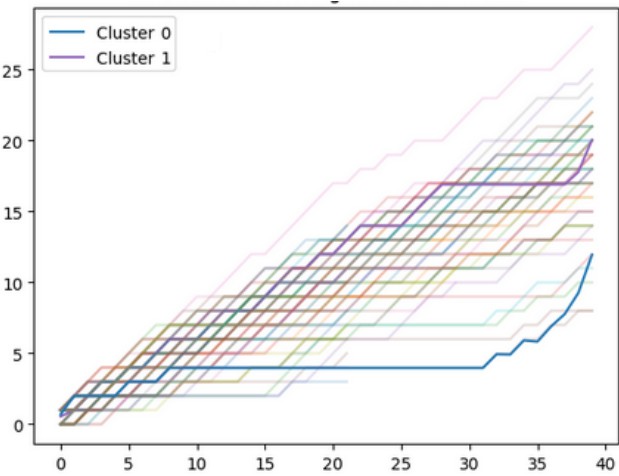

Figure 26: MAYA-Wass simulated trajectories ($\tau = 7$) clustered into two groups using the same DBA-based $k$-means procedure as for the real bees. The resulting centroids closely match those obtained from real trajectories, indicating that MAYA-Wass preserves the underlying behavioural structure captured by DBA clustering.

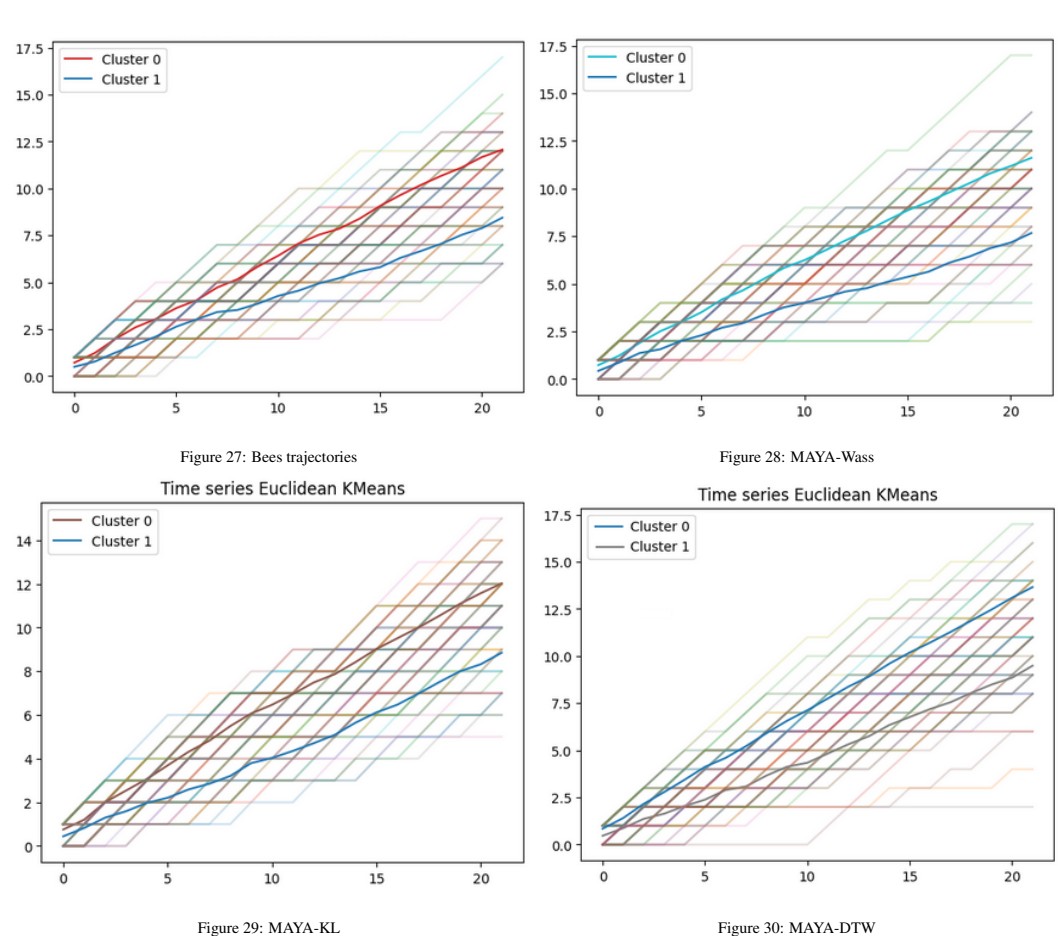

Figure 27: Bees trajectories

Figure 28: MAYA-Wass

Figure 29: MAYA-KL

Figure 30: MAYA-DTW

Figure 31: Centroïdes of two clustering of 80 bees trajectories (in Fig27) and 80 MAYA-variant (Fig28, Fig29 and Fig30) simulated trajectories (with $\tau = 7$). Clustering are done with Euclidean method (Clustering I).

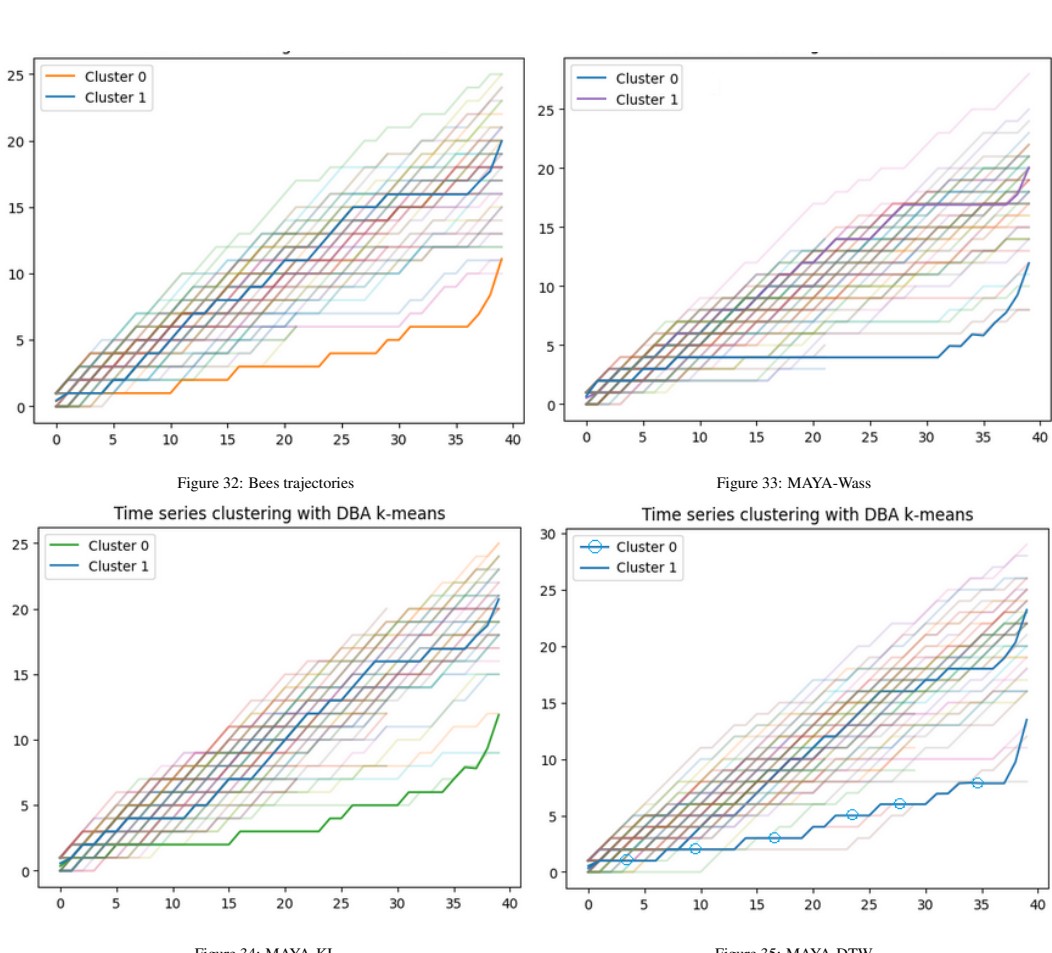

Figure 32: Bees trajectories

Figure 33: MAYA-Wass

Figure 34: MAYA-KL

Figure 35: MAYA-DTW

Figure 36: Centroïdes of two clustering of 80 bees trajectories (in Fig32) and 80 MAYA-variant (Fig33, Fig34 and Fig35) simulated trajectories (with $\tau = 7$). Clustering are done with DBA method (Clustering II).

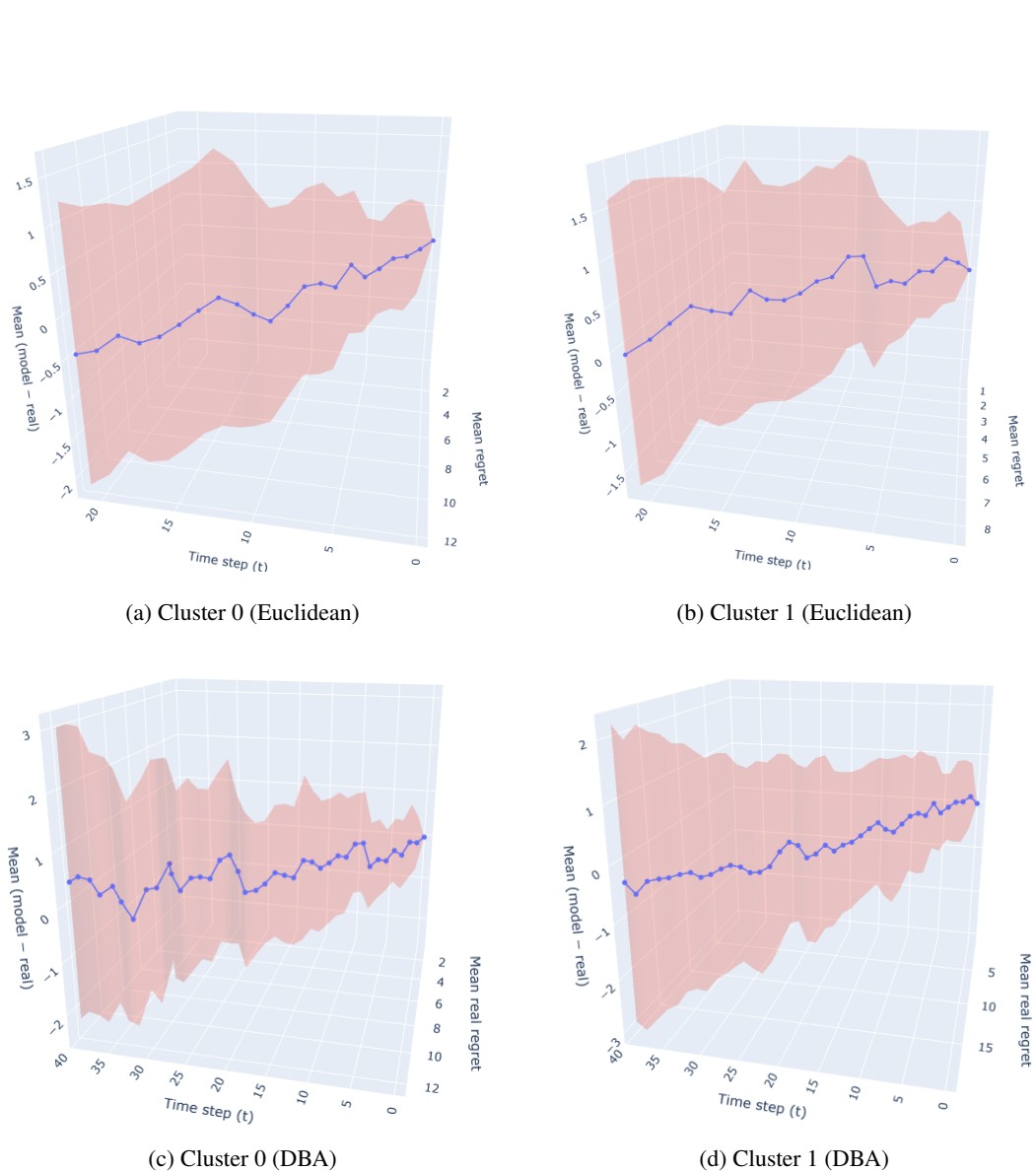

(a) Cluster 0 (Euclidean)        (b) Cluster 1 (Euclidean)

(c) Cluster 0 (DBA)        (d) Cluster 1 (DBA)

Figure 37: Average difference between MAYA-Wass ($\tau = 7$) predictions and real trajectories ($R(\pi_{\text{MAYA}}, 1, t) - R(\pi_{\text{bee}}, 1, t)$, $z$-axis) for Euclidean (top row) and DBA (bottom row) clustering, for clusters 0 and 1. The red band shows $\pm\sigma$ (standard deviation).

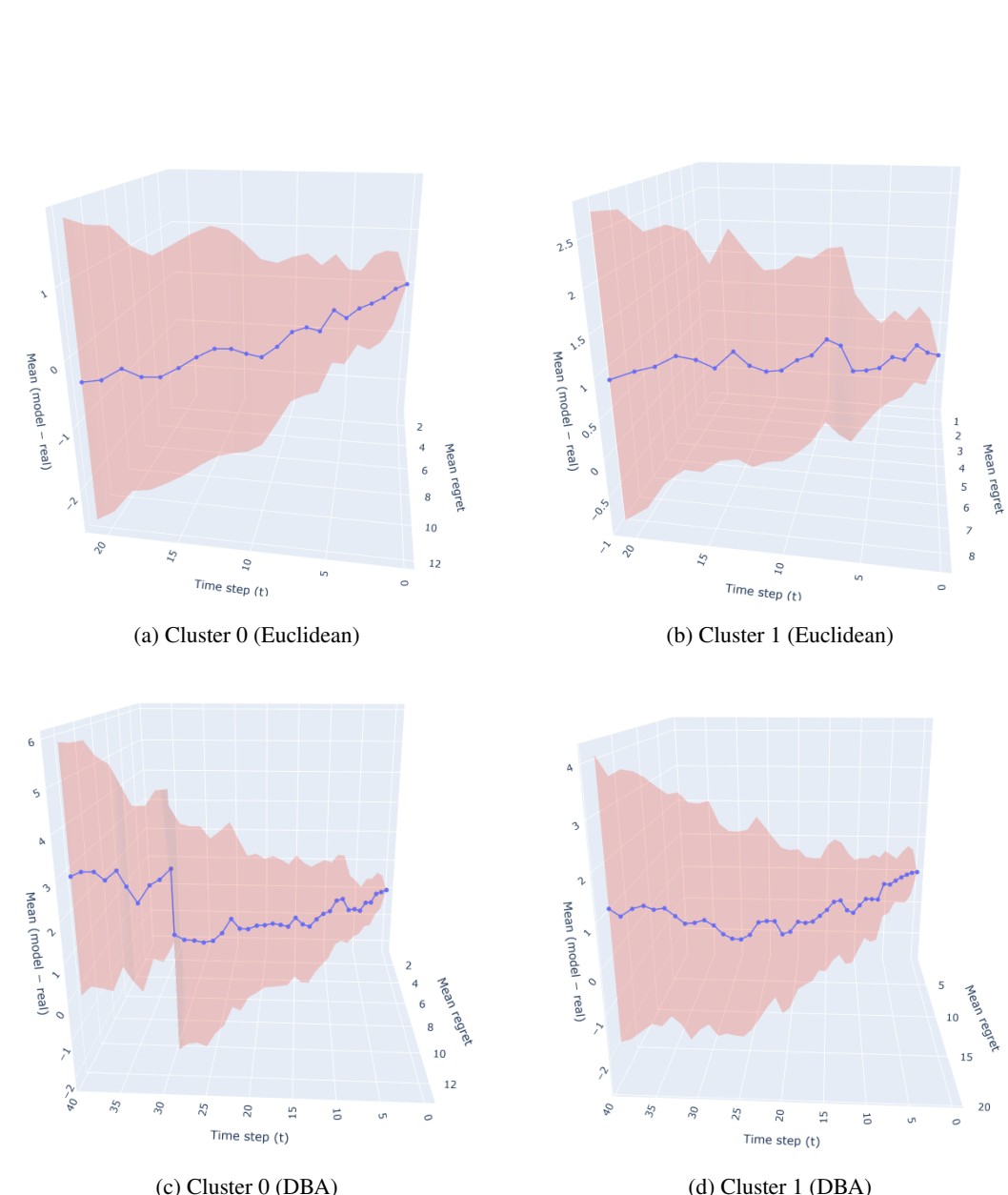

(a) Cluster 0 (Euclidean)       (b) Cluster 1 (Euclidean)

(c) Cluster 0 (DBA)       (d) Cluster 1 (DBA)

Figure 38: Average difference between MAYA-KL ($\tau = 7$) predictions and real trajectories ($R(\pi_{\text{MAYA}}, 1, t) - R(\pi_{\text{bee}}, 1, t)$, $z$-axis) for Euclidean (top row) and DBA (bottom row) clustering. The red band corresponds to $\pm\sigma$.

# 11  MAYA ALGORITHM

---

**Algorithm 1** MAYA : Multi Agent Y-maze Allocation

---

**Require:** Logged bee regret trajectory $R(\pi_{\text{bee}}, 1, T)$
**Require:** Set $\mathcal{P}$ of $N$ bandit policies $\{\pi_1, \ldots, \pi_N\}$
**Require:** Window size $\tau$ such that $t \geq \tau$
**Require:** A similarity metric $\delta$
1: $\xi = ()_{t=1}^T$
2: Init $\pi_\theta$
3: **for** $t \in \{2, \ldots, \tau - 1\}$ **do**
4:     Observe $R(\pi_{\text{bee}}, 1, t-1)$
5:     Observe a context information $x_t$
6:     **for** $i = 1$ to $N$ **do**
7:         Simulate policy agent $\pi_i(s_{t-1}|x_t)$
8:         Compute cumulative regret $R(\pi_i, 1, t-1)$
9:     **end for**
10:     $\xi_t = \text{argmin}_{\pi \in \mathcal{P}} \ \delta(\pi_{\text{bee}}, \pi, t)$
11:     $\pi_\theta(a_t|s_{t-1}) \leftarrow \pi_\xi(a_t|s_{t-1})$
12:     Select $A_t \sim \pi_\theta(a_t|s_{t-1})$
13:     Receive reward $r_t$
14:     Update $\pi_i \quad \forall \pi_i \in \mathcal{P}$
15:     $\xi[t] \leftarrow \xi_t$
16: **end for**
17: **for** $t \in \{\tau, \ldots, T\}$ **do**
18:     Observe $R(\pi_{\text{bee}}, \tau, 1, t-1)$
19:     Observe a context information $x_t$
20:     **for** $i = 1$ to $N$ **do**
21:         Simulate policy agent $\pi_i(s_{t-1}|x_t)$
22:         Compute cumulative regret $R(\pi_i, \tau, 1, t-1)$
23:     **end for**
24:     $\xi_t = \text{argmin}_{\pi \in \mathcal{P}} \ \delta(\pi_{\text{bee}}, \pi, \tau, t)$
25:     $\pi_\theta(a_t|s_{t-1}) \leftarrow \pi_\xi(a_t|s_{t-1})$
26:     Select $A_t \sim \pi_\theta(a_t|s_{t-1})$
27:     Receive reward $r_t$
28:     Update $\pi_i \quad \forall \pi_i \in \mathcal{P}$
29:     $\xi[t] \leftarrow \xi_t$
30: **end for**
31: **return** $\pi_\theta$

---

## 12 MICE DATASET EXPERIMENT

**Dataset and setup.** We use the dataset of (Ashwood et al., 2020a), which reports trial-by-trial changes in mice policy and decomposes those updates into a learning component and a noise component (see Fig. 39a). Unlike their original analysis, which simulates an average trajectory across individuals, our method (MAYA) simulates one trajectory *per* individual. The dataset contains 19 rats with between 1500 and 6000 trials each. To control the computational cost of DTW and to align with our bee experiments, we reduce the number of individual at 100.

**Selecting the memory horizon $\tau$.** According with Tab 18, Fig 39b shows MAE and MSE as a function of the memory window $\tau$. MAYA-KL clearly identifies an optimal range around $\tau \in [6, 7]$, whereas MAYA-Wass suggests $\tau \in [8, 10]$ when balancing MAE and MSE. For consistency with previous experiments, we set $\tau = 7$ in all subsequent analyses.

**Explanations and performance.** With $\tau = 7$, Fig. 48 and Fig. 44 provides MAYA explanations for the rats with the lowest and highest cumulative regret (see Fig. 40). For slow learners, all MAYA variants behave similarly (Fig. 50); for fast learners, MAYA-KL achieves the best fit, capturing rapid policy changes better than MAYA-Wass (Fig. 49). A plausible explanation is that, under KL similarity, MAYA acts more often from LinUCB-like behavior than with Wasserstein similarity (see Tab19b). As in previous datasets, MAYA-DTW tends to act more like Epsilon-Greedy, likely due to DTW's alignment properties. Overall, all MAYA variants outperform GLM baselines (Table 19a).

| side_window | MSE MAYA-KL | | MAE MAYA-KL | | MSE MAYA-Wass | | MAE MAYA-Wass | | MSE MAYA-DTW | | MAE MAYA-DTW | |
| --- | --- | --- | --- | --- | --- | --- | --- | --- | --- | --- | --- | --- |
| | mean | std | mean | std | mean | std | mean | std | mean | std | mean | std |
| 3 | 5760 | 3894 | 59 | 24 | 8083 | 5012 | 72 | 25 | 5790 | 5683 | 55 | 29 |
| 4 | 3868 | 3493 | 46 | 25 | 6547 | 3672 | 64 | 23 | 5815 | 5770 | 55 | 30 |
| 5 | 3046 | 3307 | 40 | 24 | 5724 | 3803 | 59 | 23 | 5819 | 5788 | 55 | 29 |
| 6 | 2763 | 3090 | 37 | 23 | 5276 | 3511 | 57 | 21 | 5830 | 5758 | 55 | 29 |
| 7 | 2786 | 3161 | 38 | 23 | 4640 | 3382 | 53 | 22 | 5822 | 5747 | 55 | 29 |
| 8 | 2974 | 3197 | 39 | 23 | 4728 | 3722 | 53 | 23 | 5851 | 5777 | 55 | 29 |
| 9 | 3114 | 3424 | 40 | 24 | 4231 | 3403 | 50 | 22 | 5819 | 5740 | 55 | 29 |
| 10 | 3223 | 3378 | 41 | 25 | 4197 | 3576 | 49 | 24 | 5810 | 5701 | 54 | 29 |
| 20 | 4710 | 6689 | 47 | 33 | 3491 | 3515 | 43 | 25 | 5771 | 5725 | 54 | 29 |
| 30 | 5618 | 8543 | 50 | 38 | 3453 | 3896 | 41 | 27 | 5760 | 5724 | 54 | 29 |

Table 18: MSE and MAE of MAYA as a function of the window size $\tau$ for Mice Dataset.

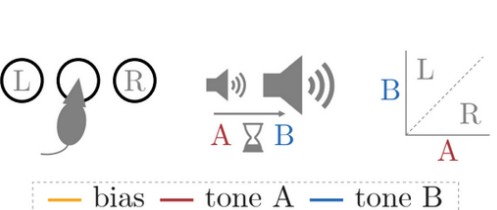

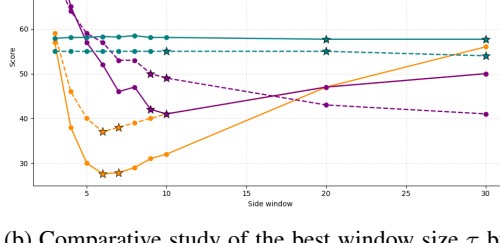

(a) According (Ashwood et al., 2020a), on each trial, a sinusoidal grating (with contrast values between 0 and 100%) appears on either the left or right side of a screen. Mice must report the side of the grating by turning a wheel (left or right) in order to receive a water reward.

(b) Comparative study of the best window size $\tau$ by average MSE and MAE. $\star$ symbol refers to the best performance according to standard deviation and average reward (see Tab.18 for the full results). MSE is displayed as $\times 10^2$.

Figure 39: Left : experimental description of the Mice Dataset. Right : Comparative study of the best window size $\tau$ for Mice Dataset.

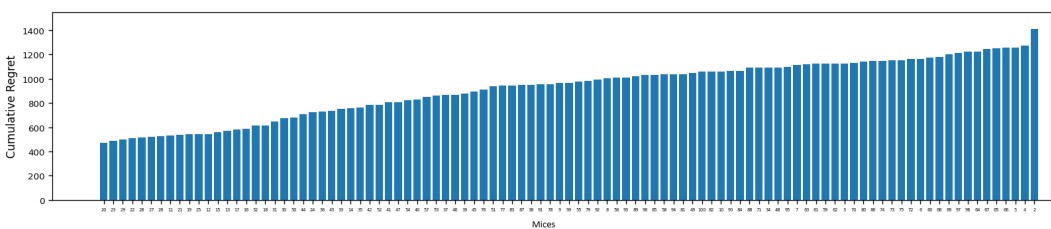

Figure 40: Proportion of cumulative regret for the Mice dataset, per mice

| | MSE | | MAE | |
|---|---|---|---|---|
| | Mean | Std | Mean | Std |
| MAYA KL | 2786 | 3161 | 38 | 23 |
| MAYA-Wass | 4640 | 3382 | 53 | 22 |
| MAYA-DTW | 5822 | 5777 | 55 | 29 |
| GLM | 6427 | 4137 | 63 | 21 |
| GLM Contextual | 6416 | 4133 | 63 | 21 |

(a)

| | Epsilon-Greedy | Lin-UCB | UCB | Uniform |
|---|---|---|---|---|
| MAYA-KL | 30%±2.5 | 2%±1.1 | 29%±1.3 | 36%±2.2 |
| MAYA-W | 27%±1.8 | 10%±1 | 28%±1 | 33%±1.5 |
| MAYA-DTW | 28%±3 | 0.5%±1 | 56%±4 | 15%±3 |

(b)

Table 19: **Left** : MSE and MAE comparison of MAYA (with $\tau = 7$ ) and GLM variants. **Right** : MAYA explainability for all MAYA choices ($\tau = 7$)

| | MAYA-KL | MAYA-Wass | MAYA-DTW |
|---|---|---|---|
| ClusterAcc (Euclidean, Max L = 1400) | 90% | 85% | 75% |
| ClusterAcc (DBA, Max L = 6000) | 80% | 75% | 65% |

Table 20: ClusterAcc (%) for Mice Datset)

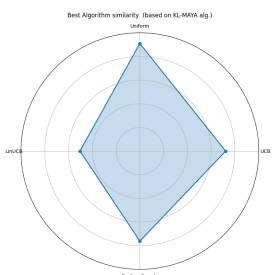
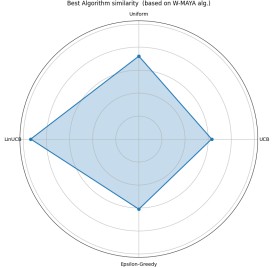
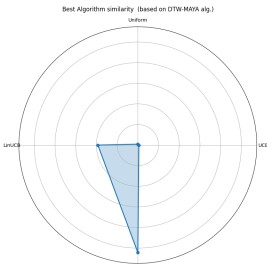

Figure 41: MAYA-KL     Figure 42: MAYA-Wass     Figure 43: MAYA-DTW

Figure 44: MAYA explainability for mouse 20 (fast learner, low regret) from Mice dataset. We report choice interpretability for MAYA-variants ($\tau = 7$). Left: LinUCB, Top: Uniform, Right: UCB, Bottom: EpsilonGreedy.

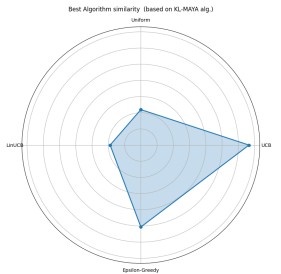
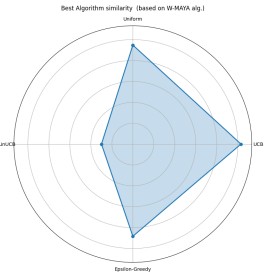
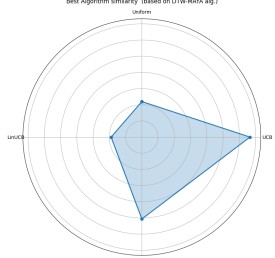

Figure 45: MAYA-KL     Figure 46: MAYA-Wass     Figure 47: MAYA-DTW

Figure 48: MAYA explainability for mouse 2 (slow learner, high regret) from Mice dataset. We report choice interpretability for MAYA-variants ($\tau = 7$). Left: LinUCB, Top: Uniform, Right: UCB, Bottom: EpsilonGreedy.

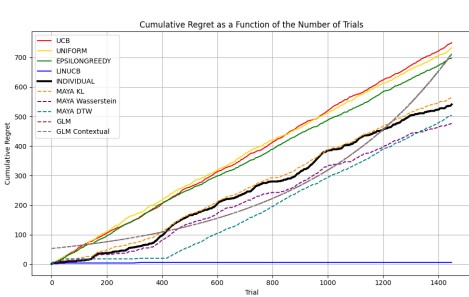

Figure 49: Mouse 20

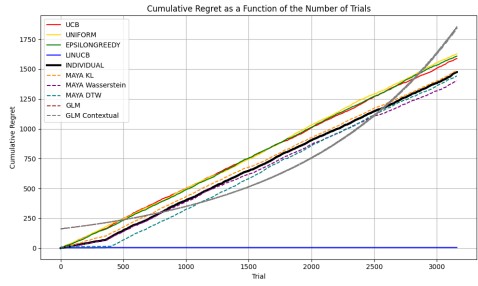

Figure 50: Mouse 2

Figure 51: Regret modelization for mouse 20 (best) and mice 2 (worst) from Mice 2, with $\tau = 7$

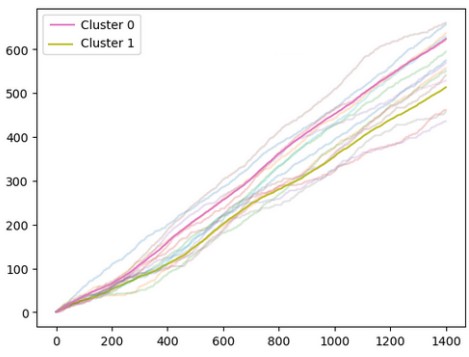

Figure 52: Mouse' trajectories

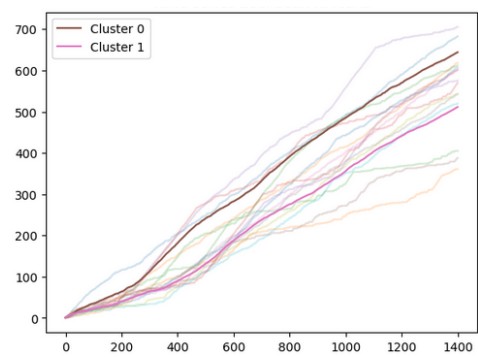

Figure 53: MAYA-KL trajectories

Figure 54: Centroides of Clustering (I) of 100 mice' (**Left**) and MAYA-KL ($\tau = 7$) (**Right**) trajectories.

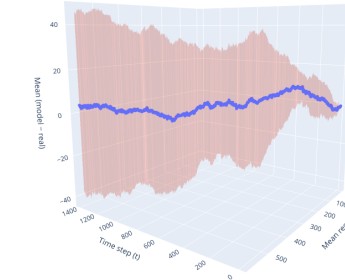

Figure 55: Cluster 0

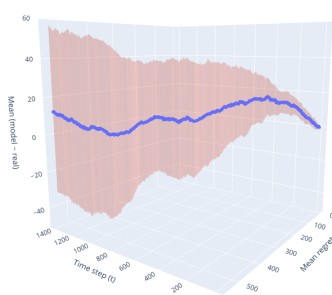

Figure 56: Cluster 1

Figure 57: Average difference between MAYA-KL ($\tau = 7$) predictions and real trajectories $(R(\pi_{\text{MAYA}}, 1, t) - R(\pi_{\text{mice}}1, t))$ (z-axis) for Euclidean (I) Clustering according 0 and 1 Cluster. Red range correspond to $\pm\sigma$ (standard deviation).

## 13 COMPLEMENTARY INFORMATION ABOUT THE BIOLOGY INTEREST

We share with other vertebrates a basic ability for abstract number representation, the *number sense* (Dehaene, 2011). As early as two days postnatally (Izard et al., 2009), this ability enables us to evaluate numbers as concepts: three books are perceived as similar to three cups, even though they differ completely in their visual features (i.e., sensory information). To evaluate quantity, both numerical and sensory information can be used. For example, when visually comparing two quantities, the larger set will often contain more items (i.e., numerosity), but may also exhibit greater density, a larger total surface area, or a wider convex hull encompassing all elements. Neuronal encoding of sensory information occurs early in the primary cortex, whereas numbers are computed in higher integrative areas by what Nieder et al. identified as *number neurons* (Nieder, 2016).

Quantity discrimination is necessary in contexts as diverse as evaluating food patches, regulating social attraction, or competing for resources (Nieder, 2020). From sharks to mammals, all major vertebrate clades appear capable of discriminating between different quantities, either spontaneously or in learning tasks (Vila Pouca et al., 2019). By carefully designing protocols that control for sensory cues, researchers have demonstrated that several non-human species are capable of performing quantity discrimination based on the abstract evaluation of numbers (Cantlon & Brannon, 2006). Among them is an insect: the honeybee (*Apis mellifera*). Beyond discriminating numerosities of up to eight items, these insects, with brains of fewer than one million neurons, can also manipulate numbers, performing simple addition, subtraction, and symbolic tasks (Dacke & Srinivasan, 2008; Gross et al., 2009; Howard et al., 2018; 2019; Giurfa et al., 2022).

Later experiments required a Y-maze: a three-armed apparatus shaped like the letter *Y*, commonly used to study memory, learning, and decision-making in rodents (Kraeuter et al., 2018) (see Fig. 58). These mazes required bees to inhibit their spatial memory (Menzel et al., 2005) (e.g., recalling that the last reward was in the left arm) and to focus instead on the visual stimuli displayed at the end of each arm. The balance between exploring new options and exploiting previously rewarded ones is key to their foraging behavior and likely plays a crucial role in their learning performance within these devices (Kembro et al., 2019; Lochner et al., 2024).

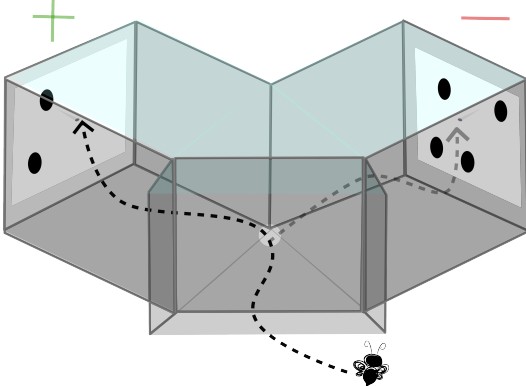

Figure 58: Y-maze for bees experiments

## 14 MATHEMATICAL PROOF OF MAYA ACCORDING TO $\tau$

**Stationary case (1) : upper bound of MAYA error**  Consider the case of two policies $\pi_1$ that achieves the highest regret i.e. $R(\pi_1, 1, T) = T$ and $\pi_0$ that achieves a zero regret i.e. $R(\pi_0, 1, T)$. In this case

$$\Delta_{\pi_1,t} - \Delta_{\pi_0,t} \leq 1 \quad \forall t$$

as the reward is in $\{0, 1\}$. The maximal bound of $R(\pi_{\text{MAYA}}, 1, T) - R(\pi_{\text{bee}}, 1, T)$ corresponds to the case where $R(\pi_{\text{bee}}, 1, T)$ is always centered between $R(\pi_1, 1, T)$ and $R(\pi_0, 1, T)$ (see Fig59a). Let's define $\varepsilon_t^*$ the agent who act the closest of the bee at $t$ and $\varepsilon_t$ the agent chosen by MAYA at $t$. Then

$$\mathbb{P}[\varepsilon_t = \varepsilon_t^*] = 0.5 \quad \forall t$$

as no best agent are better from the other one. This case corresponds to an equality between the two possible agent (with extreme regret values) and leads to the worst scenario of a stationary case when the similarity distance $d()$ are when define. Then the maximal cumulative gap between MAYA-regret and Bee-regret in stationary case are :

$$\sum_{t=1}^{T} |\Delta_{\text{MAYA},t} - \Delta_{\text{Bee},t}| \leq \frac{1}{2} \sum_{t=1}^{T} |\Delta_{\pi_1,t} - \Delta_{\text{Bee},t}| + \frac{1}{2} \sum_{t=1}^{T} |\Delta_{\pi_0,t} - \Delta_{\text{Bee},t}|$$

$$\leq \sum_{t=1}^{T} \frac{t}{2}$$

$$\leq \frac{\frac{T}{2}(\frac{T}{2} + 1)}{2}$$

$$\leq \frac{1}{8}(T(T + 2)) \tag{1}$$

**Stationary case (2) : upper bound of the worst policy**  Consider the case where $\pi_{\text{MAYA}}$ always chose like $\pi_1$ and $\pi_{\text{bee}}$ always chose like $\pi_0$ (see Fig 59b). Then the similarity distance $d()$ fails to provide a correct measure and MAYA chose the agent with the largest regret gap relative to the bee's regret. Then for all $t$

$$\mathbb{P}[\varepsilon_t \neq \varepsilon_t^*] = 1.$$

Then the maximal cumulative gap between MAYA-regret and Bee-regret in the worst policy in stationary case are :

$$\sum_{t=1}^{T} |\Delta_{\text{MAYA},t} - \Delta_{\text{Bee},t}| \leq \sum_{t=1}^{T} |\Delta_{\pi_1,t} - \Delta_{\pi_0,t}|$$

$$\leq \frac{T \cdot (T + 1)}{2} \tag{2}$$

The alternative case where $\pi_{\text{MAYA}}$ always chooses as $\pi_0$ and $\pi_{\text{bee}}$ always chooses as $\pi_1$ is equivalent.

**Cyclic case : upper bound of MAYA error with no windows ($\tau = T$) policy**  Consider that after $S$ trials the bee moves from $\pi_1$ to $\pi_0$ (alternative cases are equivalent, see Fig 60a). Consider that the distances are well defined, as in the stationary case (1). Then :

$$\sum_{t=1}^{S} |\Delta_{\text{MAYA},t} - \Delta_{\text{Bee},t}| \leq \frac{1}{8}(S \times (S + 2)) \tag{3}$$

The time required for MAYA to act like $\pi_0$ is $2S + 1$ but at $t = 2S + 1$, the bee changes from $\pi_0$ to $\pi_1$ and MAYA continues to act like $\pi_1$ (see Fig.60a). Recursively, MAYA always act like $\pi_1$ from $t = 1$ until $t = T$. Then

$$\mathbb{P}[\varepsilon_t = \pi_1] = 1 \quad \forall t$$

and

$$\mathbb{P}[\varepsilon_t = \varepsilon_t^*] = \frac{N_*(T)}{T}, \quad \forall t$$

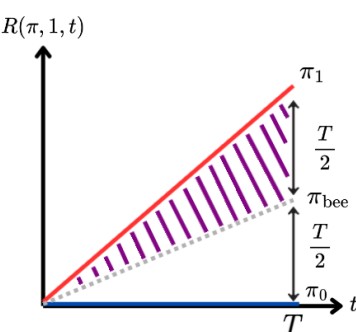
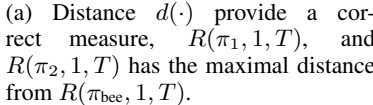
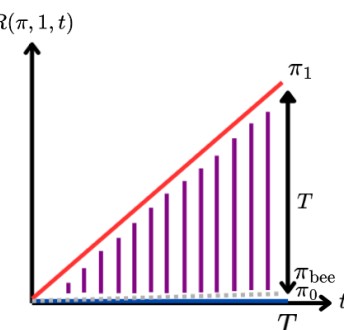

(a) Distance $d(\cdot)$ provide a correct measure, $R(\pi_1, 1, T)$, and $R(\pi_2, 1, T)$ has the maximal distance from $R(\pi_{\text{bee}}, 1, T)$.

(b) Distance $d(\cdot)$ fails to provide a correct measure. MAYA alawys selects actions as the agent whose behavior is farthest from that of the bee.

Figure 59: Maximal cumulative gap between MAYA-regret and Bee-regret in **stationary case** according the distance $d(\cdot)$ abilities to provide a correct measure

Where

$$N_*(T) = qS + \min(S, r),$$

$$q = \left\lfloor \frac{T}{2S} \right\rfloor,$$

$$r = T - 2Sq \in [0, 2S).$$

A minimal bound of $N_*$ are :

$$N_*(T) \geq \frac{T}{2}$$

Then the maximal cumulative gap between MAYA-regret and Bee-regret in a cyclic case with no windows is :

$$\sum_{t=1}^{T} |\Delta_{\text{MAYA},t} - \Delta_{\text{Bee},t}| \leq \frac{N_*(T)}{T} \frac{1}{8}(T.(T+2)) + (1 - \frac{N_*(T)}{T}) \frac{T.(T+1)}{2}$$

$$\leq \frac{T}{2} \frac{1}{T} \frac{1}{8}(T.(T+2)) + (1 - \frac{T}{2} \frac{1}{T}) \frac{T.(T+1)}{2}$$

$$= \frac{T(5T+6)}{16} \tag{4}$$

**Cyclic case : upper bound of MAYA error with windows $\tau = S$**  Assume that $S$ are even. Consider that after $S$ trials, the bee moves from $\pi_1$ to $\pi_0$ (alternative cases are equivalent, see Fig60b). Consider that the distance is well define like in the stationary case (1). From time $t = 1$ until $S$, MAYA act as the best agent :

$$\sum_{t=1}^{S} |\Delta_{\text{MAYA},t} - \Delta_{\text{Bee},t}| \leq \frac{1}{8}(S \times (S+2)) \tag{5}$$

and

$$\mathbb{P}[\varepsilon_t = \varepsilon_t^*] = 1 \quad \forall t \in \{1, \ldots, S\}.$$

From time $S + 1$ until $S + \frac{S}{2}$, MAYA acts as the worst policy (start cycle)

$$\sum_{t=S+1}^{S+\frac{S}{2}} |\Delta_{\text{MAYA},t} - \Delta_{\text{Bee},t}| \leq \sum_{t=S+1}^{S+\frac{S}{2}} t \tag{6}$$

$$\leq \frac{S(5S+2)}{8} \tag{7}$$

and

$$\mathbb{P}[\varepsilon_t \neq \varepsilon_t^*] = 1 \quad \forall t \in \{S+1, \dots, S + \frac{S}{2}\}.$$

And from $t = S + \frac{S}{2} + 1$ until $t = 2S$ MAYA acts with the best policy (end cycle):

$$\sum_{t=S+\frac{S}{2}+1}^{2S} |\Delta_{\text{MAYA},t} - \Delta_{\text{Bee},t}| \leq \sum_{t=S+\frac{S}{2}+1}^{2S} \frac{t}{2}$$

$$\leq \frac{S(7S+2)}{16} \tag{8}$$

and

$$\mathbb{P}[\varepsilon_t = \varepsilon_t^*] = 1 \quad \forall t \in \{S + \frac{S}{2} + 1, \dots, 2S\}.$$

Consider a full cycle, the event $\varepsilon_t = \varepsilon_t^*$ appears $S - \frac{S}{2}$ times. Let's set

$$q = \left\lfloor \frac{\max(0, T-S)}{S} \right\rfloor, \qquad r = \max(0, T-S) - qS \in [0, S).$$

Here $q$ is the number of full cycle $S$ in $t > S$, and $r$ is the rest of a potential unfinished tail segment of the started cycle. Let $N_*(T) = \sum_{t=1}^{T} 1_{\varepsilon_t = \varepsilon^*}$ with $N_*(T) \leq T$ equal to

$$N_*(T) = \min(T, S) + q \cdot \frac{S}{2} + \max(0, r - \frac{S}{2})$$

If $S$ is even and $T > S$ then

$$N_*(T) \geq \frac{T}{2} + \frac{S}{4} \tag{9}$$

*Proof:*
*With $T = S + qS + r$:*

$$N_*(T) - \left(\frac{T}{2} + \frac{S}{4}\right) = \frac{S}{2} - \frac{r}{2} + \max(0, r - \frac{S}{2}) \geq 0,$$

*where the minimum are archived with $r = \frac{S}{2}$.*

$$\mathbb{P}[\varepsilon_t = \varepsilon_t^*] = \frac{N_*(T)}{T} \geq \frac{1}{2} + \frac{S}{4T} \tag{10}$$

In the cases where $S$ is not not even

$$q = \left\lfloor \frac{T-S}{S} \right\rfloor, \qquad r = T - S - qS \in [0, S).$$

then

$$N_*(T) = S + \frac{q(S+1)}{2} + \max\left(0, r - \frac{S-1}{2}\right).$$

As $T = S + qS + r$, we have

$$N_*(T) - \frac{T}{2} = \frac{S}{2} + \frac{q}{2} + \max\left(0, r - \frac{S-1}{2}\right) - \frac{r}{2}.$$

and for any $r \in [0, S)$,

$$\min_r \left(\max(0, r - \frac{S-1}{2}) - \frac{r}{2}\right) = -\frac{S-1}{4}.$$

Then

$$N_*(T) \geq \frac{S}{2} + \frac{q}{2} - \frac{S-1}{4} + \frac{T}{2} = \frac{S+1}{4} + \frac{q}{2} + \frac{T}{2} \geq \frac{S+1}{4} + \frac{T}{2}.$$

$$N_*(T) \geq \frac{T}{2} + \frac{S+1}{4} \geq \frac{T}{2} + \frac{S}{4}. \tag{11}$$

Which are better to the $S$ parity case.

Then the maximal cumulative gap between MAYA-regret and Bee-regret with windows $\tau = S$ is

$$\sum_{t=1}^{T} |\Delta_{\text{MAYA},t} - \Delta_{\text{Bee},t}| \le \frac{N_*(T)}{T} \frac{T(T+2)}{8} + (1 - \frac{N_*(T)}{T}) \frac{T(T+1)}{2}$$

$$\le (\frac{T}{2} + \frac{S}{4}) . \frac{1}{T} . \frac{T(T+2)}{8} + (1 - (\frac{T}{2} + \frac{S}{4}) . \frac{1}{T}) \frac{T(T+1)}{2}$$

$$\le \frac{10T^2 + 12T - 3ST - 2ST}{32} \tag{12}$$

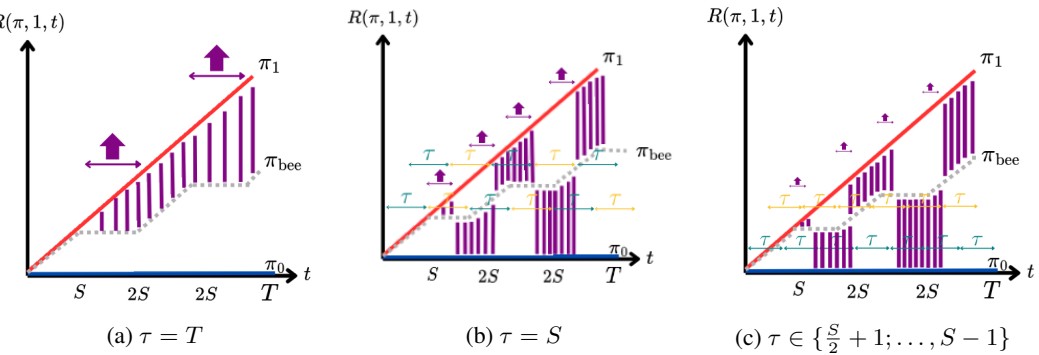

(a) $\tau = T$       (b) $\tau = S$       (c) $\tau \in \{\frac{S}{2}+1; \ldots, S-1\}$

Figure 60: Maximal cumulative gap between MAYA regret and bee regret in a non-stationary case, measured with respect to window $\tau$. The purple arrow highlights the period during which MAYA chooses actions in accordance with the agent whose behavior is most distant from that of the bee.

**Cyclic case : upper bound of MAYA error with windows** $\tau \in \{\frac{S}{2}+1; \ldots, S-1\}$ . We consider the case where $\frac{S}{2} + 1 \le \tau < S$ (see Fig60c). Assume that $S$ are even. From time $t = 1$ until $S$, MAYA act as the best agent (stationary case 1) :

$$\sum_{t=1}^{S} |\Delta_{\text{MAYA},t} - \Delta_{\text{Bee},t}| \le \frac{1}{8}(S \times (S+2)) \tag{13}$$

and

$$\mathbb{P}[\varepsilon_t = \varepsilon^*] = 1 \quad \forall t \in \{1, \ldots, S\}.$$

From time $S + 1$ until $S + \frac{\tau}{2}$, MAYA acts as the worst policy (start cycle)

$$\sum_{t=S+1}^{S+\frac{\tau}{2}} |\Delta_{\text{MAYA},t} - \Delta_{\text{Bee},t}| \le \sum_{t=S+1}^{S+\frac{\tau}{2}} t$$

$$\le \frac{\tau}{4}(2S + 1 + \frac{\tau}{2})$$

$$\le \frac{\tau^2}{8} + \frac{S\tau}{2} + \frac{\tau}{4} \tag{14}$$

and

$$\mathbb{P}[\varepsilon_t \ne \varepsilon^*] = 1 \quad \forall t \in \{S+1, \ldots, S+\frac{\tau}{2}\}.$$

And from $t = S + \frac{\tau}{2} + 1$ until $t = 2S$, MAYA acts as the best policy (end cycle) with :

$$\sum_{t=S+\frac{\tau}{2}+1}^{2S} |\Delta_{\text{MAYA},t} - \Delta_{\text{Bee},t}| \le \sum_{t=S+\frac{\tau}{2}+1}^{2S} \frac{t}{2}$$

$$\le \frac{(3S + \frac{\tau}{2} + 1)(S - \frac{\tau}{2})}{4} \tag{15}$$

and

$$\mathbb{P}[\varepsilon_t = \varepsilon_t^*] = 1 \quad \forall t \in \{S + \frac{\tau}{2} + 1, \dots, 2S\}.$$

Consider a full cycle, the event $\varepsilon_t = \varepsilon_t^*$ appears $S - \frac{\tau}{2}$ times. Let's set

$$q = \lfloor \frac{T - S}{S} \rfloor \qquad r = (T - S) - qS \in [0, S).$$

Let $N_*(T) = \sum_{t=1}^{T} 1_{\varepsilon_t = \varepsilon^*}$ with $N_*(T) \leq T$ equal to

$$N_*(T) = S + q(S - \frac{\tau}{2}) + \max(0, r - \frac{\tau}{2}).$$

and

$$\mathbb{P}[\varepsilon_t = \varepsilon_t^*] = \frac{N_*(T)}{T} \tag{16}$$

The maximal cumulative gap between MAYA-regret and Bee-regret with windows $\tau \in \{\frac{S}{2} + 1; \dots, S - 1\}$ with $S$ parity is

$$\sum_{t=1}^{T} |\Delta_{\text{MAYA},t} - \Delta_{\text{Bee},t}| \leq \frac{N_*(T)}{T} \cdot \frac{T(T+2)}{8} + \left(1 - \frac{N_*(T)}{T}\right) \cdot \frac{T(T+1)}{2}$$

$$\leq \frac{S + q(S - \frac{\tau}{2}) + \max(0, r - \frac{\tau}{2}).}{T} \cdot \frac{T(T+2)}{8}$$
$$+ (1 - \frac{S + q(S - \frac{\tau}{2}) + \max(0, r - \frac{\tau}{2}).}{T}) \cdot \frac{T(T+1)}{2}$$

As $N_*(T) \geq T(1 - \frac{\tau}{2S})$ without any condition on $S$ parity, the maximal cumulative gap between the MAYA-regret and the Bee-regret with windows $\tau \in \{\frac{S}{2} + 1; \dots, S - 1\}$ is

$$\sum_{t=1}^{T} |\Delta_{\text{MAYA},t} - \Delta_{\text{Bee},t}| \leq \frac{T(T+2)}{8} + \frac{(3T+2)T}{16} \frac{\tau}{S} \tag{17}$$

**Cyclic case : upper bound of MAYA with windows** $\tau < \frac{S}{2} + 1$    In this case, there is no way to be sure that the distance $d()$ do not fails to identify the best agent. It's equivalent to choose randomly and the worst case corresponds to the upper bound of the worst policy. Then the maximal cumulative gap between MAYA regret and Bee-regret with $\tau < \frac{S}{2} + 1$ in cyclic case are equivalent to Eq. 2.

**Cyclic case : upper bound of MAYA with windows** $\tau > S$    In this case, the time required to change the policy is over a cycle $S > 1$. Then, the bee switch two times in $\tau$ and MAYA allows it to act as the same agent. Then it is equivalent to act as a cyclic case with no windows ($\tau = T$) Then the maximal cumulative gap between MAYA regret and Bee-regret with $\tau > S$ in cyclic case are equivalent to Eq. 4.

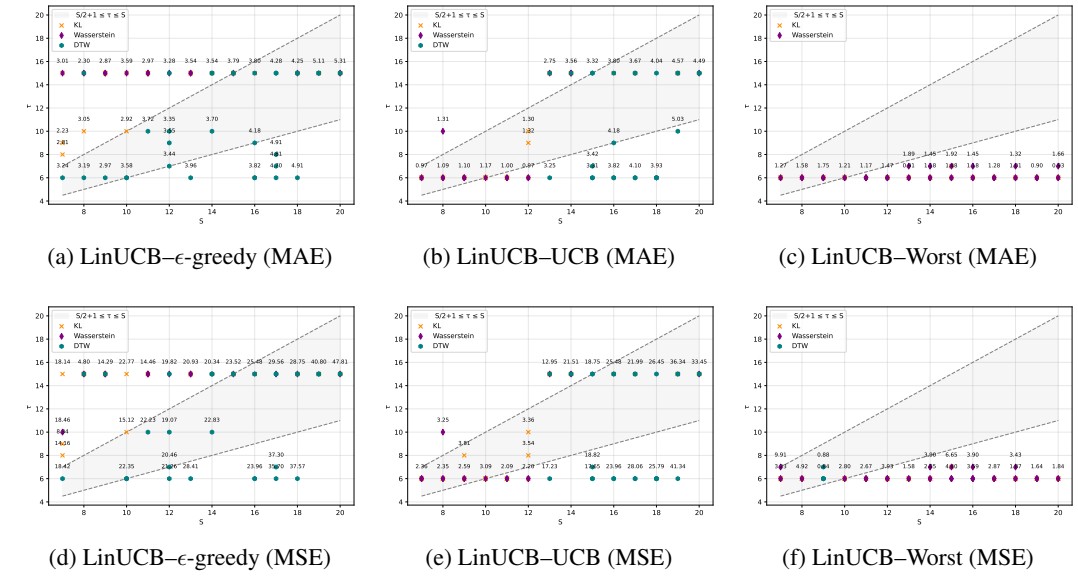

Figure 61: Top–5 minimal MAE/MSE values according to the memory parameter $\tau$ under the three similarity metrics (KL, Wasserstein, DTW) for the simulated bee trajectories with (unknown by MAYA) switching period $S$. Each column corresponds to a policy-shift after $S$ trials scenario (LinUCB–$\epsilon$-greedy, LinUCB–UCB, LinUCB–Worst), and each row reports either MAE or MSE scoring. Dashed lines show the theoretical constraint $\frac{S}{2} + 1 \le \tau \le S$.

## 15 SIMULATIONS

We extended our experimental protocol to include 42 simulated datasets, resulting in more than 100.800 synthetic trajectories. These trajectories were generated by introducing controlled policy shifts (after $S$ trials) between two bandit strategies that are observed in the real data and interest biologist (e.g., LinUCB $\leftrightarrow$ UCB and LinUCB $\leftrightarrow$ $\epsilon$-greedy). We additionally simulated a 'Lin-UCB $\leftrightarrow$ Worst" condition, i.e., a switch from the best-performing to the worst-performing policy, in order to stress-test MAYA under extreme behavioural changes.

Across all these simulated scenarios (see Fig. 61), our results confirm that constraining the memory parameter to the interval

$$\frac{S}{2} + 1 \;\le\; \tau \;\le\; S$$

leads to a consistent minimization of the loss between the full cumulative-regret trajectory of the simulated bee and the cumulative-regret trajectory produced by MAYA. The only exception arises in the "LinUCB $\leftrightarrow$ Worst" condition, where we can set $\tau < \frac{S}{2}$. This behaviour is expected: the two policies are extremely different and nearly stationary (their rewards are almost always equal to 1 or 0, respectively), so a very small memory window is sufficient to discriminate their cumulative regret sequence. Overall, these analyses validate the theoretical justification of our $\tau$-range and demonstrate its empirical robustness across diverse switching regimes.

**Grid search over bandit parameters** ($\alpha_{\mathbf{UCB}}, \alpha_{\mathbf{LinUCB}}, \epsilon$   We provide here a small grid search over exploration parameters using held-out simulated policy. For $\epsilon$-greedy we tested $\epsilon \in \{0.1, 0.2, 0.3\}$, and for UCB and LinUCB we used $\alpha_{\mathrm{ucb}}, \alpha_{\mathrm{linucb}} \in \{0.5, 1, 1.5, 2, 4\}$.

Our results show that the value of $\alpha_{\mathrm{linucb}}$ has only a limited effect, because rewards in our task are a fully deterministic function of the context (the arm with the largest stimulus, encoded in the integer-valued context vector). Increasing either $\epsilon$ or $\alpha_{\mathrm{ucb}}$ effectively drives the corresponding policy toward uniform exploration, which in turn increases $c_t$ if the behavior of the bee are none in a uniform style and reduces the diversity of candidate policies.

For LinUCB and UCB, we set the exploration parameter to $\alpha = 1$, a standard default value for binary rewards in contextual bandit implementations (Li et al., 2010; Bouneffouf & Claeys, 2021); typical

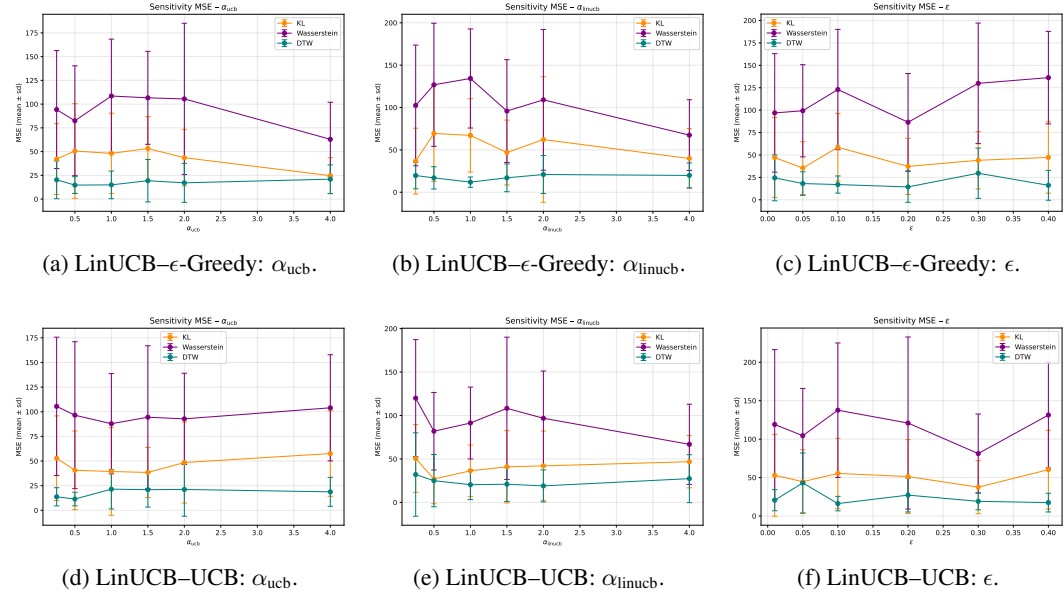

(a) LinUCB–$\epsilon$-Greedy: $\alpha_{\text{ucb}}$.    (b) LinUCB–$\epsilon$-Greedy: $\alpha_{\text{linucb}}$.    (c) LinUCB–$\epsilon$-Greedy: $\epsilon$.

(d) LinUCB–UCB: $\alpha_{\text{ucb}}$.    (e) LinUCB–UCB: $\alpha_{\text{linucb}}$.    (f) LinUCB–UCB: $\epsilon$.

Figure 62: MSE Sensitivity analysis of MAYA across three hyperparameters ($\alpha_{\text{ucb}}$, $\alpha_{\text{linucb}}$, $\epsilon$) and two simulated bees policies (Shift LinUCB–$\epsilon$-Greedy and Shift LinUCB–UCB). We fix $S = 12$ and $\tau = 7$ for generate simulations.

LinUCB libraries set $\alpha = 1$ by default). For e-greedy, we use $\epsilon = 0.2$ (20% random exploration), a conventional choice in empirical bandit studies with Bernoulli rewards (e.g., standard tutorials and empirical evaluations typically consider $\epsilon = [0.1, 0.3]$)

## 16 EXPLANATION OF THE SIMILARITY METRICS

The *instantaneous regret* at trial $t$ is defined as $\Delta_t = r(s_t, a_t^\star) - r(s_t, a_t)$, where $a_t^\star := \pi^\star(s_t) = \arg\max_{a \in \mathcal{A}} r(s_t, a)$ is the optimal action under the state $s_t$. The *cumulative simple regret* after $T$ trials is the sum of instantaneous regrets $R(\pi, 1, T) = \sum_{t=1}^T \Delta_{\pi,t}$.

At each trial $t$, the bee selects an action $a_t^{\text{bee}}$ and MAYA selects $a_t^{\text{MAYA}}$. We define the *cost of reproduction cost*

$$c_t := c(s_t \mid a_t^{\text{MAYA}}) = \begin{cases} 1, & \text{if } a_t^{\text{MAYA}} \neq a_t^{\text{bee}}, \\ 0, & \text{otherwise.} \end{cases}$$

The sequence of $c_t$ is the binary vector (start from the first trial 1 until $T$).

$$\mathbf{c} = (c_1, \ldots, c_T),$$

and the cumulative reproduction-cost trajectory $C_T = \sum_{t=1}^T c_t$ is used in the MSE/MAE evaluation.

This quantity is closely related to classical simple regret. For any policy $\pi$,

$$\Delta_{\pi,t} = r(s_t, a_t^\star) - r(s_t, a_t), \qquad a_t^\star = \arg\max_{a \in \mathcal{A}} r(s_t, a),$$

and the cumulative simple cumulative regret is

$$R(\pi, 1, T) = \sum_{t=1}^T \Delta_{\pi,t}.$$

Therefore, the cumulative difference in regret after $t$ trials is bounded by the number of disagreements:

$$\left| R(\pi_{\text{bee}}, 1, t) - R(\pi_{\text{MAYA}}, 1, t) \right| \leq \sum_{s=1}^t c_s$$

Intuitively, the two regrets can only drift apart on trials where the bee and MAYA disagree, and each such case can increase their regret difference by at most one unit.

We recall that as this is related to a full deterministic experiment, there is no stochasticity in the reward function ( as it always related to the Y-maze side with the highest number of stimuli)

Then :

$$C_t = \sum_{i=1}^t c_t$$

The evaluation metrics reported in the paper are:

$$\text{MSE}(C_T), \qquad \text{MAE}(C_T),$$

according the setting (KL/Wasserstein/DTW distances) of MAYA.

**Worked numeric example.** Consider $T = 5$ trials. Suppose the bee obtains reward 1 when selecting $L$ and reward 0 when selecting $R$. Then:

| $t$ | $a_t^{\text{bee}}$ | $a_t^{\text{MAYA}}$ | $c_t$ | $C_t$ | $R(\pi_{\text{bee}}, 1, t)$ | $R(\pi_{\text{MAYA}}, 1, t)$ | $\left\| R(\pi_{\text{bee}}, 1, t) - R(\pi_{\text{MAYA}}, 1, t) \right\|$ |
|---|---|---|---|---|---|---|---|
| 1 | $L$ | $L$ | 0 | 0 | 0 | 0 | 0 |
| 2 | $L$ | $R$ | 1 | 1 | 0 | 1 | 1 |
| 3 | $R$ | $R$ | 0 | 1 | 1 | 1 | 0 |
| 4 | $L$ | $R$ | 1 | 2 | 1 | 2 | 1 |
| 5 | $L$ | $L$ | 0 | 2 | 1 | 2 | 1 |

Thus

$$\mathbf{c} = (0, 1, 0, 1, 0), \qquad (C_t)_{t=1}^5 = (0, 1, 1, 2, 2),$$

and

$$R(\pi_{\text{bee}}, 1, T) = (0, 0, 1, 1, 1), \qquad R(\pi_{\text{MAYA}}, 1, T) = (0, 1, 1, 2, 2).$$

The mean squared error between the cumulative reproduction cost and the bee's ideal trajectory is:

$$\text{MSE} = \frac{1}{5}(0^2 + 1^2 + 1^2 + 2^2 + 2^2) = 1.8.$$

This illustrates the exact sequences used by the similarity metrics in MAYA. Since the cumulative disagreement $C_t$ upper-bounds the difference in cumulative regret between the bee and the model,

$$\left| R(\pi_{\text{bee}}, 1, t) - R(\pi_{\text{MAYA}}, 1, t) \right| \leq C_t,$$

the process $C_t$ provides a direct and interpretable proxy for trajectories divergence. Studying its mean squared error (MSE) and mean absolute error (MAE) between the bee and the model therefore offers complementary insight into the quality of imitation. While trajectory-wise regret metrics capture global differences in learning performance, the MSE and MAE of $C_t$ quantify how tightly the model reproduces the *pattern of action choices* over time. Low MSE/MAE values indicate that the model not only matches the scale of regret but also closely follows the trial-by-trial structure of action agreements and disagreements, providing a finer-grained measure of imitation quality.

## 17    DISCLOSURE OF LLM USE

Large Language Models (LLMs) were used in a limited capacity during the preparation of this paper. Their use was restricted to (i) spelling and phrasing assistance (to support a dyslexic co-author), and (ii) suggesting improvements to Python scripts for graph generation and visualization. No part of the scientific content, analyses, or conclusions was produced by LLMs.

