{S}{4}) \cdot \frac{1}{T} \cdot \frac{T(T+2)}{8} + (1 - (\frac{T}{2} + \frac{S}{4}) \cdot \frac{1}{T})\frac{T(T+1)}{2} \\
&\leq \frac{10T^2 + 12T - 3ST - 2ST}{32}
\end{aligned}
\tag{12}
$$

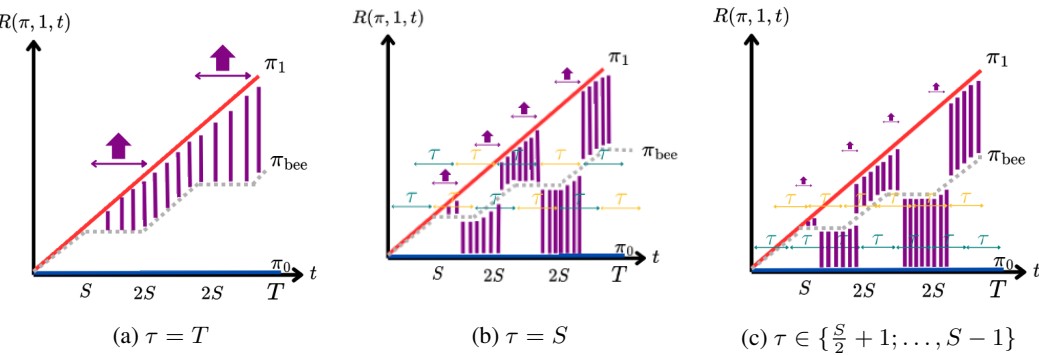

(a) $\tau = T$        (b) $\tau = S$        (c) $\tau \in \{\frac{S}{2}+1; \ldots, S-1\}$

Figure 75: Maximal cumulative gap between MAYA regret and bee regret in a non-stationary case, measured with respect to window $\tau$. The purple arrow highlights the period during which MAYA chooses actions in accordance with the agent whose behavior is most distant from that of the bee.

**Cyclic case : upper bound of MAYA error with windows** $\tau \in \{\frac{S}{2}+1; \ldots, S-1\}$ . We consider the case where $\frac{S}{2} + 1 \leq \tau < S$ (see Fig75c). Assume that $S$ are even. From time $t = 1$ until $S$, MAYA act as the best agent (stationary case 1) :

$$
\sum_{t=1}^{S} |\Delta_{\text{MAYA},t} - \Delta_{\text{Bee},t}| \leq \frac{1}{8}(S \times (S+2))
\tag{13}
$$

and

$$
\mathbb{P}[\varepsilon_t = \varepsilon^*] = 1 \quad \forall t \in \{1, \ldots, S\}.
$$

From time $S+1$ until $S + \frac{\tau}{2}$, MAYA acts as the worst policy (start cycle)

$$
\begin{aligned}
\sum_{t=S+1}^{S+\frac{\tau}{2}} |\Delta_{\text{MAYA},t} - \Delta_{\text{Bee},t}| &\leq \sum_{t=S+1}^{S+\frac{\tau}{2}} t \\
&\leq \frac{\tau}{4}(2S + 1 + \frac{\tau}{2}) \\
&\leq \frac{\tau^2}{8} + \frac{S\tau}{2} + \frac{\tau}{4}
\end{aligned}
\tag{14}
$$

and

$$
\mathbb{P}[\varepsilon_t \neq \varepsilon^*] = 1 \quad \forall t \in \{S+1, \ldots, S+\frac{\tau}{2}\}.
$$

And from $t = S + \frac{\tau}{2} + 1$ until $t = 2S$, MAYA acts as the best policy (end cycle) with :

$$
\begin{aligned}
\sum_{t=S+\frac{\tau}{2}+1}^{2S} |\Delta_{\text{MAYA},t} - \Delta_{\text{Bee},t}| &\leq \sum_{t=S+\frac{\tau}{2}+1}^{2S} \frac{t}{2} \\
&\leq \frac{(3S + \frac{\tau}{2} + 1)(S - \frac{\tau}{2})}{4}
\end{aligned}
\tag{15}
$$

and
$$\mathbb{P}[\varepsilon_t = \varepsilon_t^*] = 1 \quad \forall t \in \{S + \frac{\tau}{2} + 1, \ldots, 2S\}.$$

Consider a full cycle, the event $\varepsilon_t = \varepsilon_t^*$ appears $S - \frac{\tau}{2}$ times. Let's set

$$q = \lfloor \frac{T-S}{S} \rfloor \qquad r = (T-S) - qS \in [0, S).$$

Let $N_*(T) = \sum_{t=1}^T 1_{\varepsilon_t = \varepsilon^*}$ with $N_*(T) \leq T$ equal to

$$N_*(T) = S + q(S - \frac{\tau}{2}) + \max(0, r - \frac{\tau}{2}).$$

and

$$\mathbb{P}[\varepsilon_t = \varepsilon_t^*] = \frac{N_*(T)}{T} \tag{16}$$

The maximal cumulative gap between MAYA-regret and Bee-regret with windows $\tau \in \{\frac{S}{2} + 1; \ldots, S-1\}$ with $S$ parity is

$$\sum_{t=1}^T |\Delta_{\text{MAYA},t} - \Delta_{\text{Bee},t}| \leq \frac{N_*(T)}{T} \cdot \frac{T(T+2)}{8} + \left(1 - \frac{N_*(T)}{T}\right) \cdot \frac{T(T+1)}{2}$$

$$\leq \frac{S + q(S - \frac{\tau}{2}) + \max(0, r - \frac{\tau}{2}).}{T} \cdot \frac{T(T+2)}{8}$$

$$+ (1 - \frac{S + q(S - \frac{\tau}{2}) + \max(0, r - \frac{\tau}{2}).}{T}) \cdot \frac{T(T+1)}{2}$$

As $N_*(T) \geq T(1 - \frac{\tau}{2S})$ without any condition on $S$ parity, the maximal cumulative gap between the MAYA-regret and the Bee-regret with windows $\tau \in \{\frac{S}{2} + 1; \ldots, S-1\}$ is

$$\sum_{t=1}^T |\Delta_{\text{MAYA},t} - \Delta_{\text{Bee},t}| \leq \frac{T(T+2)}{8} + \frac{(3T+2)T}{16} \frac{\tau}{S} \tag{17}$$

**Cyclic case : upper bound of MAYA with windows $\tau < \frac{S}{2} + 1$**   In this case, there is no way to be sure that the distance $d()$ do not fails to identify the best agent. It's equivalent to choose randomly and the worst case corresponds to the upper bound of the worst policy. Then the maximal cumulative gap between MAYA regret and Bee-regret with $\tau < \frac{S}{2} + 1$ in cyclic case are equivalent to Eq. 2.

**Cyclic case : upper bound of MAYA with windows $\tau > S$**   In this case, the time required to change the policy is over a cycle $S > 1$. Then, the bee switch two times in $\tau$ and MAYA allows it to act as the same agent. Then it is equivalent to act as a cyclic case with no windows ($\tau = T$) Then the maximal cumulative gap between MAYA regret and Bee-regret with $\tau > S$ in cyclic case are equivalent to Eq. 4.

## 15 DISCLOSURE OF LLM USE

Large Language Models (LLMs) were used in a limited capacity during the preparation of this paper. Their use was restricted to (i) spelling and phrasing assistance (to support a dyslexic co-author), and (ii) suggesting improvements to Python scripts for graph generation and visualization. No part of the scientific content, analyses, or conclusions was produced by LLMs.