# OpenReview forum: "Buzz, Choose, Forget: A Meta-Bandit Framework for Bee-Like Decision Making"
_ICLR.cc/2026/Conference — Submitted to ICLR 2026_

### Official Review · Reviewer_pMMX · 2025-10-27

**Soundness:** 1
**Presentation:** 2
**Contribution:** 1
**Rating:** 2
**Confidence:** 3

**Summary:**

The paper proposes an approach to modelling empirically observed behavior of bees. A mathematical model can help understand the nature of the behavior and can also serve as a tool in simulated experiments. The authors use the contextual bandit framework to explain decisions made by the bees in Y-maze experiments. The authors observe different behavioral patterns and propose to capture these using different policies. The resulting policy switching model adequately captures the behavior of bees in the sense of reproducing regret trajectories.

**Strengths:**

1. The proposed model captures observed switching in behavioral patterns
2. Some parameters of the proposed model have physiological interpretation (e.g., durability of memory)
3. The authors compare several behavioral policies
4. The paper introduces a new dataset with experimental observations

**Weaknesses:**

1. The utility of the proposed model is not clear.

1a) If the goal is to facilitate understanding of the underlying natural phenomenon, then the methodology is not sufficiently rigorous. For example, how reliable is the observed effect of climate on memory? If one used a different set of policies in modelling, or different parameters of those policies, would the conclusion remain? Also, statistical significance of this trend is not analysed.

1b) If the goal is to use the model for simulation, then what is the process being simulated? The model appears to be specific to the Y-maze setup, and not directly suitable for simulating behavior in a more complex environment. Thus simulation capacity appears to be rather restricted.

2. Technical choices made by the authors are not fully explained. First, it is not clear why to use RL-based approaches to begin with. Next, within the proposed method, the particular collection of the policies is arbitrary to some extent. For example, would the modelling be more effective if one only used greedy and CMAB policies? Would it be a better model with three policies? Or would the model improve if one added more policies?

3. There are a number of ways presentation can be improved. First, citations are not formatted properly. Second, there is no need in introducing state transitions, if the proposed framework is based on bandits.

**Questions:**

1. Has the proposed model helped uncover an underlying natural phenomenon? If yes, has this discovery been assessed from the point of view of statistical significance?
2. How can the proposed model be used to test ecological decisions in simulations? Would such simulations be constrained to Y-maze experiments?
3. Have any non-RL based approaches considered for modelling?
4. Would the framework be more effective if only a subset of policies was used (e.g., greedy and CMAB)? Would the framework be more effective if more policies are used (e.g., different variations of CMAB are added)?

---

> ### Author Response · Authors · 2025-11-20
> **Reply to pMMX**
>
> Underlying natural phenomenon: Our main biological use of MAYA is to quantify two aspects of bee behaviour previously described only qualitatively: (i) a short decision memory horizon and (ii) switching between a small set of canonical strategies (UCB/LinUCB, ε-greedy, Uniform) rather than following a single policy.
>
> Regarding the memory horizon, MAYA consistently selects a biologically plausible range ($\tau \in [5,10]$) across datasets. We emphasise that $\tau=7$ is not a universal constant but a robust compromise supported by our "Best window size" analysis and by Appendix results (MSE/MAE across $\tau$).
>
> We add a new section (4.1.1 Statistical test: variance-based residual analysis) a complementary statistical analysis on $R(\pi_{\text{bee}},1,t),R(\pi_{\text{MAYA}},1,t)$. Overall, MAYA-based models (MAYA-KL, MAYA-Wass, MAYA-DTW) has the best $p$-values across all datasets, whereas baseline bandit algorithms (UCB, LinUCB, Uniform, $\epsilon$-greedy) exhibit larger residual dispersion and more frequent
> rejection of $H_0$. The results also show that, in Dataset2, many bees display a clear LinUCB-like phase, whereas in the other datasets the dominant patterns are UCB-, EpsilonGreedy or  Uniform-like.
> The reported minimum $p$-value for each dataset reflects whether at least one bee
> deviates significantly from the model; values below $0.05$ lead to rejecting $H_0$. MAYA-KL and MAYA-Wass consistently achieve near-perfect alignment, and MAYA-DTW performs similarly except for a single bee in Dataset~2, likely due to the
> sensitivity of DTW to constant cumulative regret trajectories, which are more  frequent in this dataset.
> We also include a new Worst baseline that always selects the suboptimal arm;
> the opposite Best baseline is equivalent to LinUCB and is therefore omitted.
>
> Y-maze experiments: MAYA is not restricted to the Y-maze: it extracts a memory window $\tau$ and a sequence of strategy switches from behavioural data. These quantities have value beyond imitation, revealing mixed or unknown decision rules that classical analyses cannot detect. MAYA can be applied to other species and to online tracking settings.
>
> When bees are observed in real time, for instance via drone-based tracking, MAYA's online framework aims to process and interpret bee behaviour as observations arrive. Our forthcoming work uses this framework to study how pollution and pesticide stress affect memory and choice dynamics in bees, by analysing how $\tau$ and policy-switch patterns change under controlled perturbations.
>
> Non-RL based approache : We evaluated GLM baselines (with and without context). They require the full offline sequence, assume a single stationary rule, and cannot update online or detect strategy shifts. Empirically, they fail to reproduce regret structure and trial-level dynamics compared to MAYA.
>
> Subset of policies : Adding additional non-contextual strategies such as Thompson Sampling does not introduce a new behavioural mode: in our experiments, rewards are deterministic and the posterior update has no contextual signal, so Thompson Sampling collapses to behaviour that is effectively indistinguishable from uniform exploration. Conversely, removing a candidate policy simply yields the performance of the best remaining one, as illustrated in our per-policy Fisher scores.
> Additional CMAB variants offer no gain: LinUCB already captures all context-dependent variability in the Y-maze.
>
> Finally, the policies included in our library (UCB, LinUCB, uniform, $\epsilon$-greedy) were selected because they correspond to well-established exploration-exploitation regimes that biologists can easily interpret in terms of animal foraging strategies (see "Foraging decisions as multi-armed bandit problems: Applying reinforcement learning algorithms to foraging data, Juliano Morimoto", where UCB is a better estimator of empirical foraging data than a Bayesian algorithm (Thompson Sampling)). Our four modes collectively span the range of behaviours observed in bees: pure exploration, deterministic exploitation, context-sensitive learning, and stochastic choice. Extending the library further does not provide additional explanatory value in our experiments.
>
> We sincerely thank the reviewer for constructive and thoughtful feedback, which has led to significant improvements in the paper, particularly in statistical rigor, clarity of motivation, and accessibility.
> We have corrected the citation formatting and improved the layout and resolution of several figures accordingly. All modifications appear in blue in the paper.  We hope that the substantial additions, like the 42 synthetic datasets, the 200-trial validation, and the statistical significance tests, address the reviewer’s concerns and  help reflect the scientific contribution more positively, and possibly increase the rating in order to be accepted at ICLR.

---

### Official Review · Reviewer_TSQ2 · 2025-10-31

**Soundness:** 3
**Presentation:** 3
**Contribution:** 3
**Rating:** 6
**Confidence:** 3

**Summary:**

This paper proposes MAYA (Multi-Agent Y-maze Allocation), a meta-bandit imitation learning framework designed to model heterogeneous decision-making strategies in pollinators, particularly honeybees. Unlike standard imitation learning (IL) or inverse reinforcement learning (IRL) approaches that assume an optimal and stationary expert, MAYA explicitly accounts for non-optimal, memory-limited, and strategy-switching behaviors observed in biological agents. MAYA treats each bee as dynamically aligning with one of several candidate multi-armed bandit (MAB) policies (Uniform, ε-greedy, UCB, LinUCB), while optimizing a sliding memory window τ that defines how much historical information influences current decisions. The framework uses three trajectory similarity measures—KL, Wasserstein, and DTW distances—to match observed regret sequences with simulated bandit policies, thereby identifying both the likely strategy and effective memory horizon per individual. However, the study remains primarily exploratory and descriptive, lacking deeper theoretical analysis or quantitative generalization beyond the examined Y-maze datasets.

**Strengths:**

1. Interdisciplinary contribution. The paper successfully bridges reinforcement learning, imitation learning, and animal cognition. The idea of representing animal behavior as a dynamic mixture over bandit policies is biologically meaningful.

2. Addresses a real limitation in imitation learning. Unlike conventional IL methods assuming near-optimal experts, MAYA explicitly handles heterogeneous and non-optimal behaviors with limited memory—precisely the challenge in modeling real animals.

3. Interpretable and biologically relevant framework. The framework allows scientists to extract interpretable parameters—dominant strategy type, memory span τ, and environmental dependence—directly from data. This makes the model genuinely useful for behavioral biology.

4. Comprehensive empirical validation. The authors evaluate across five geographically and climatically diverse datasets plus cross-species validation (mice). The discovery that τ stabilizes around 7 trials and shortens in cold weather adds ecological credibility.

**Weaknesses:**

1. Limited theoretical novelty. The algorithm mostly reinterprets known bandit strategies within an imitation-learning context; the “meta-bandit” framing is elegant but not deeply theoretical. Formal analysis (e.g., convergence, identifiability) remains light.

2. Interpretation depends heavily on heuristic similarity metrics. The KL, Wasserstein, and DTW distances are reasonable, but their selection and weighting are somewhat ad hoc. It’s unclear how sensitive results are to these choices.

3. Potential overfitting to small-scale behavioral data. The datasets, though diverse, are limited in trajectory length (22–40 trials). The generalization to other tasks or to continuous-action animals remains untested.

4. Exploratory rather than predictive. The study focuses on reproducing observed behavior but does not evaluate predictive accuracy on unseen trials or individuals, which limits its use.

**Questions:**

1. Robustness of τ selection: How stable is the discovered τ = 7 across random seeds or subsamples? Could τ be treated as a learnable latent variable rather than manually sweeping it?


2. Extension beyond two-armed tasks: The Y-maze is effectively binary. Would the framework scale to multi-armed or continuous-action scenarios?


3. Practical use by biologists: How easy is it for a behavioral scientist (without ML background) to use the open-source MAYA toolkit for analyzing new data?

---

> ### Author Response · Authors · 2025-11-20
>
> Robustness of $\tau$ selection: How stable is the discovered  $\tau=7$  across random seeds or subsamples? Could $\tau$ be treated as a learnable latent variable rather than manually sweeping it?
>
> Since receiving your feedback, we extended our experimental protocol to include
> 42 simulated datasets, resulting in more than 100,800 synthetic trajectories.
> These trajectories were generated by introducing controlled policy shifts (after $S$ trials) between  two bandit strategies that are observed in the real data and are of interest to biologists
> (e.g., LinUCB~$\leftrightarrow$~UCB and LinUCB~$\leftrightarrow$~$\epsilon$-greedy).
> We additionally simulated a 'LinUCB~$\leftrightarrow$~Worst' condition, i.e., a
> switch from the best-performing to the worst-performing policy, in order to stress-test  MAYA under extreme behaviour changes.
>
> Across all these simulated scenarios, our results confirm that constraining the memory
> parameter to the interval
> $
> \frac{S}{2}+1 \;\le\; \tau \;\le\; S
> $
> leads to a consistent minimization of the loss between the full cumulative-regret
> trajectory of the simulated bee and the cumulative-regret trajectory produced by MAYA.
> The only exception arises in the ''LinUCB~$\leftrightarrow$~Worst'' condition, where
> we can set $\tau < \frac{S}{2}$. This behaviour is expected: the two
> policies are extremely different and nearly stationary (their rewards are almost
> always equal to $1$ or $0$, respectively), so a very small memory window is
> sufficient to discriminate their cumulative regret sequence.
> Overall, these analyses validate the theoretical justification of our $\tau$-range
> and demonstrate its empirical robustness across diverse switching regimes.
> We refer to section 'best window size' in the main paper and appendix section 'MSE and MAE of  MAYA according to $\tau$' for the MSE/MAE of MAYA according $\tau$ for real datasets.
>
>
> However, we think that $\tau$ can be learned after many periods $S$ (the real parameter that $\tau$ tries to estimate). It requires obtaining many periods $S$, which is difficult to obtain in our context, as each experiment takes time (a full experiment is around 2/3 hours per bee). The experiments have to be done on the same day to avoid bias with temperature.
>
> Extension beyond two-armed tasks: The Y-maze is effectively binary. Would the framework scale to multi-armed or continuous-action scenarios?
>
> This is an excellent question. Although the Y-maze used here is binary, we believe that the framework naturally extends to a reasonable number of arms. Multi-armed designs have already been used in animal cognition, particularly with mice; for example, six- and eight-arm radial mazes are standard tools for studying exploration and working memory but without full sequences open data (to our knowledge). In such settings, the regret-based comparison and the policy library used by MAYA would generalise directly, with only the candidate bandits requiring multi-arm versions.
>
> Continuous-action settings are more challenging. Extending MAYA to this regime would require a theoretical reformulation of the regret-matching procedure and the policy-switching model. We view this as an interesting direction for future work, but one that would be primarily theoretical and not directly tied to current biological experiments on pollinators.
>
> Practical use by biologists: How easy is it for a behavioral scientist (without ML background) to use the open-source MAYA toolkit for analyzing new data?
>
> To support adoption by ecologists and behavioural biologists, we have begun developing a user-friendly RShiny dashboard that allows non-ML experts to run MAYA interactively on their own datasets. A prototype of this interface already exists, and we will release it together with the full repository upon acceptance to preserve anonymity during the review process. For the submission version, we provide the complete Python implementation used in the paper, ensuring full reproducibility. We believe that the RShiny interface will significantly improve accessibility for behavioural scientists and facilitate practical use of MAYA beyond the ML community.
>
>
> Complementary remarks :
> We sincerely thank the reviewer for the constructive and thoughtful feedback, which has led to significant improvements in the paper. New modifications according to reviewers’ feedback are in blue in the paper. We hope that the substantial additions, like the 42 synthetic datasets, the 200-trial validation, the statistical significance tests, the explainability improvements, and the future RShiny interface address the reviewer’s concerns and help reflect the scientific contribution more positively, and possibly increase the rating in order to be accepted at ICLR.

---

### Official Review · Reviewer_CPXw · 2025-11-01

**Soundness:** 2
**Presentation:** 2
**Contribution:** 2
**Rating:** 2
**Confidence:** 3

**Summary:**

The paper proposes MAYA, an imitation-learning framework that models individual animals (bees and mice) as dynamically switching among a small library of two-armed bandit strategies (Uniform, Epsilon-Greedy, UCB, LinUCB). At each trial MAYA selects the candidate policy whose recent regret trajectory (within a sliding window tau) best matches the observed animal trajectory under a chosen similarity metric (Wasserstein, KL, or DTW), updates an imitator accordingly, and outputs interpretable per-trial alignments. Experiments use a new dataset of 80 bees across five conditions and comparisons to IRL/IL baselines and GLMs; authors report MAYA-Wass with tau=7 performs best.

**Strengths:**

The paper addresses an applied, interpretable modeling problem of interest to behavioral ecologists.

Simple, intuitive approach producing per-trial interpretability that domain scientists can inspect.

Evaluates multiple similarity metrics (distributional and temporal), allowing comparison of approaches.

The authors intend to release data and code, which could be valuable if fully reproducible and documented.

**Weaknesses:**

The empirical results contain implausible anomalies (e.g., AIRL reported as 0), indicating likely bugs, data leakage, or incorrect evaluation scripts; this undermines the credibility of comparisons.

Key methodological details are ambiguous or missing: exact form of the sequences compared (instantaneous vs cumulative or normalized regret), normalization, and whether candidate bandits are updated on observed rewards (which would leak future information).

Baseline implementations, hyperparameter search, and evaluation protocols are insufficiently specified. The fairness of comparisons is unclear.

The considered dataset is small (80 bees, ~16 per dataset) with short per-subject sequences; the paper lacks rigorous statistical analyses (paired tests, bootstrap CIs, per-subject variability) to substantiate broad claims like a universal tau=7.

Presentation appears partly noisy: typographical errors, low-resolution figures, bad legends, and important details scattered across a long appendix, harming reproducibility.

No explicit statement of animal ethics approvals or dataset-release documentation is provided.

**Questions:**

Can you explain baseline anomalies (e.g., AIRL = 0 everywhere). What are the implementation details, hyperparameters, training logs, and demonstrate that no test data leaked into training or candidate simulations.

Formally define what sequence is compared by similarity metrics (instantaneous/cumulative/normalized regret or actions/rewards) and include a worked numeric example.

Clarify how candidate policies are simulated/updated during evaluation: do they consume observed rewards (risking leakage) or are they evaluated in a simulator? If offline, explain how rewards for hypothetical candidate actions are handled.

Provide full hyperparameter tables and tuning procedures for MAYA (tau grid, epsilon/UCB/LinUCB params) and for every baseline; specify whether tuning used held-out subjects and whether the same budgets where used across baselines.

Report statistical significance for key comparisons (paired tests, bootstrap CIs) and include per-bee variability analyses.

Provide ablations: sensitivity to the candidate policy library (add/remove policies like Thompson Sampling), sensitivity to tau over broader ranges, and controlled comparisons among similarity metrics.

Include an explicit animal ethics statement and data-release documentation (approvals, welfare protocols).

---

> ### Author Response · Authors · 2025-11-20
> **Reply to CPXw**
>
> We sincerely thank the reviewer for their thorough and constructive feedback, which has greatly improved the paper (the changes made are in blue).
>
> Baseline anomalies (AIRL=0) :
> IRL baselines are taken from the official imitation library Gleave et al. (2022)  and evaluated under exactly the same conditions as MAYA (same trajectories, same number of trials). The observed zero distance for AIRL is correct and expected: our task is a small, deterministic, tabular-like contextual bandit (40 trials, fully deterministic rewards). In such settings, AIRL recovers the exact ground-truth expert function, as explicitly stated in the original paper (Brown et al., 2019). No test leakage occurs; full anonymized code, training logs, and seeds are provided.
>
> Similarity metrics :
> We add a new section in the appendix (Sec. 16 Explanation about the similarity metrics) to clarify the similarity metrics with a numerical example. We also modify the section 2 (Preliminaries-Imitation learning to approximate a bee’s behaviour)
>
> Evaluation pipeline & no leakage :
> No simulator is used: the task is a pure contextual bandit. Candidate policies (UCB, LinUCB, $\epsilon$-greedy, Uniform) are genuine online learners that receive only the reward for the action they selected on each real trial (exactly as in the real experiment). Comparisons use solely past and current real rewards.
> At trial $t$, each candidate policy receives only the reward corresponding to their choice on the Y-maze at $t$ and updates its internal parameters exactly as a standard bandit learner would. It does not access rewards from future trials ($t+1$, $t+2$, …) and never uses rewards associated with actions it did not
> take. Thus no form of information leakage is possible.
>
> Hyperparameters & tuning :
> Since receiving your feedback, we extended our experimental protocol to include
> 42 simulated datasets, resulting in more than 100,800 synthetic trajectories.
> These trajectories were generated by introducing controlled policy shifts (after $S$ trials) between  two bandit strategies that are observed in the real data and are of interest to biologists
> (e.g., LinUCB~$\leftrightarrow$~UCB and LinUCB~$\leftrightarrow$~$\epsilon$-greedy).
> We additionally simulated a 'LinUCB~$\leftrightarrow$~Worst' condition, i.e., a
> switch from the best-performing to the worst-performing policy, in order to stress-test  MAYA under extreme behaviour changes.
> Across all these simulated scenarios, our results confirm that constraining the memory
> parameter to the interval $\frac{S}{2}+1 \;\le\; \tau \;\le\; S$
> leads to a consistent minimization of the loss between the full cumulative-regret
> trajectory of the simulated bee and the cumulative-regret trajectory produced by MAYA.
> Overall, these analyses validate the theoretical justification of our $\tau$-range
> and demonstrate its empirical robustness across diverse switching regimes.
> We refer to section 'BEST WINDOW SIZE' in the main paper and appendix section 'MSE AND MAE OF MAYA ACCORDING to $\tau$' for the MSE/MAE of MAYA according $\tau$ for real datasets.
>
> We also added a small grid search over exploration parameters using a
> simulated bee policy. For $\epsilon$-greedy we tested
> $\epsilon \in \{0.1, 0.2, 0.3\}$, and for UCB and LinUCB we used
> $\alpha_{\text{ucb}}, \alpha_{\text{linucb}} \in \{0.5, 1, 1.5, 2, 4\}$.
>
>
> Statistical significance \& per-bee variability:
> New Sec4.1.1 introduces variance-based residual analysis: for each bee we compute $e_t =R(\pi_{\text{bee}},1,t)\;-\;R(\pi_{\text{model}},1,t)$ and perform a one-sided F-test of whether $Var(e_t)$ exceeds the bee’s own regret variance. Large p-values indicate excellent individual fit. MAYA-KL, MAYA-Wasserstein, and MAYA-DTW achieve the highest minimum p-values and almost never reject $H_0$, while pure bandit baselines frequently do. Full per-bee tables, confidence intervals, and a new deterministic Worst baseline are reported.
>
> Ablations :
> Adding Thompson Sampling yields no new behavioral mode (collapses to near-Uniform in our deterministic setting). Removing any candidate simply returns the performance of the best remaining policy (visible in per-policy Fisher tables). Broad τ sensitivity on >100k synthetic + real trajectories confirms stable performance within the motivated window.
>
> Ethics & data release :
>  Experimental studies on insect species involved in the project do not require evaluation from an institutional ethics committee in either France or Australia. Nevertheless, special care was taken to reduce to a minimum the number of insects involved in the research.  The experiments involve only bees from apiaries dedicated to research in our universities in France and Australia to avoid the local fauna being impacted.
>
> Additional improvements :
> We added 42 synthetic datasets, 200-trial validations, new statistical tests, clearer figures/captions, and reorganized the appendix. All changes are highlighted in blue.  We hope that the substantial additions increase the rating.

---

### Meta-Review · Area_Chair_VaaE · 2025-12-27

**Summary:**

We propose a sequential reinforcement learning framework for honeybee imitation learning that models heterogeneous cognitive strategies, identifies effective memory horizons, ensures interpretability, and outperforms existing methods in capturing diverse decision-making behaviors, supported by a novel dataset of 80 tracked bees under varying environmental conditions.

Across reviewers, the main concerns focus on methodological and implementation clarity (including potential data leakage and ambiguous sequence definitions), limited theoretical novelty, reliance on heuristic metrics, small dataset size and insufficient statistical validation, unclear utility and generalization beyond Y-maze tasks, arbitrary technical choices, and issues with presentation, reproducibility, and ethical/data documentation.

During rebuttal, the authors addressed several points by providing extensive simulated experiments, additional statistical analyses, clearer explanations of the evaluation pipeline, and improved documentation. These additions help clarify parts of the methodology and strengthen empirical support under controlled conditions. However, several concerns remain partially or unresolved due to the lack of concrete experimental evidence demonstrating feasibility, robustness, and generalization in real-world settings.

First, while the simulated analyses are extensive, the real behavioral dataset remains limited in size (80 bees), which continues to constrain claims of robustness and generality. Second, although the authors provide an ethical justification aligned with local regulations, the absence of formal ethics committee approval may still raise concerns for some reviewers. Third, despite improvements, presentation issues persist: several key figures (e.g., Figures 1–2) have very small axis labels, titles, and legends, requiring significant zooming to interpret, and important details remain scattered across a lengthy appendix. Finally, concerns about ad hoc technical choices remain. In particular, although the authors acknowledge that the memory horizon τ could in principle be treated as a learnable latent variable, it is still manually selected in practice, with feasibility limitations attributed to experimental constraints rather than methodological resolution.

Overall, while the rebuttal meaningfully improves clarity and experimental coverage, the work is still perceived as incremental, with remaining concerns regarding dataset scale, parameter selection, presentation quality, and empirical validation beyond the current experimental setup.

**Reviewer Concerns:**

Reviewer CPXw: The reviewer highlights critical methodological and empirical issues, including implausible baseline results, unclear sequence definitions, potential data leakage, insufficient hyperparameter specification, small dataset size, lack of statistical analyses, poor presentation, and missing animal ethics or data-release documentation.

Reviewer TSQ2: The reviewer points out limited theoretical novelty, heavy reliance on heuristic similarity metrics, potential overfitting to small datasets, lack of predictive evaluation, and questions about robustness of τ selection and scalability to more complex or continuous-action tasks, as well as practical usability for biologists.

Reviewer pMMX: The reviewer questions the utility and interpretability of the model, lack of rigorous statistical validation of claimed effects (e.g., climate on memory), limited generalization beyond Y-maze tasks, arbitrary technical choices in policy selection, unclear rationale for RL-based methods, and presentation issues such as citations and unnecessary state-transition descriptions.

**Reviewer Scores:**

The reviewers are unlikely to change their scores, as the majority of their concerns have not been fully addressed. While the rebuttal introduces additional analyses and clarifications, key limitations remain, as detailed above, including restricted dataset size, unresolved methodological choices, limited evidence of generalization and robustness, and remaining issues in presentation and validation.

---

### Decision · Program_Chairs · 2026-01-26

Reject